# A tuft cell - ILC2 signaling circuit provides therapeutic targets to inhibit gastric metaplasia and tumor development

Ryan N. O'Keefe [1,2], Annalisa L. E. Carli [1,2], David Baloyan[1,2], David Chisanga [1,2], Wei Shi[1,2], Shoukat Afshar-Sterle[1,2], Moritz F. Eissmann [1,2], Ashleigh R. Poh [1,2], Bhupinder Pal[1,2], Cyril Seillet [3,4], Richard M. Locksley [5,6], Matthias Ernst [1,2,7] & Michael Buchert [1,2,7] ✉

Although gastric cancer is a leading cause of cancer-related deaths, systemic treatment strategies remain scarce. Here, we report the pro-tumorigenic properties of the crosstalk between intestinal tuft cells and type 2 innate lymphoid cells (ILC2) that is evolutionarily optimized for epithelial remodeling in response to helminth infection. We demonstrate that tuft cell-derived interleukin 25 (IL25) drives ILC2 activation, inducing the release of IL13 and promoting epithelial tuft cell hyperplasia. While the resulting tuft cell - ILC2 feed-forward circuit promotes gastric metaplasia and tumor formation, genetic depletion of tuft cells or ILC2s, or therapeutic targeting of IL13 or IL25 alleviates these pathologies in mice. In gastric cancer patients, tuft cell and ILC2 gene signatures predict worsening survival in intestinal-type gastric cancer where ~40% of the corresponding cancers show enriched co-existence of tuft cells and ILC2s. Our findings suggest a role for ILC2 and tuft cells, along with their associated cytokine IL13 and IL25 as gatekeepers and enablers of metaplastic transformation and gastric tumorigenesis, thereby providing an opportunity to therapeutically inhibit early-stage gastric cancer through repurposing antibody-mediated therapies.

In this work we demonstrate that there is a feedforward circuit between tuft cells and ILC2s that promotes gastric metaplasia and cancer. Indeed, by treating with inhibiting antibodies that target tuft cell-ILC2 interactions it is possible to inhibit tumor growth, as such this circuit could pose as a therapeutic target for the treatment of gastric cancer.

Gastric cancer (GC) is the 3rd leading cause of cancer-related mortality worldwide with a predicted increase of 40% over the next two decades[1,2], on top of the current one million new cases annually and close to 900,000 deaths[2,3], thereby emphasising the need for

greater understanding of molecular drivers of the disease. In addition to genetic, environmental and lifestyle factors, the risk of developing GC[4,5] is associated with chronic gastric metaplasia, a frequent non-malignant pre-neoplastic precursor to GC[6] that occurs from chronic gastritis. The latter arises as a consequence of persistent bacterial infection[7] and is characterized by epithelial tissue remodeling, loss of gastric acid-secreting parietal cells, and the expansion of metaplastic cells that resemble intestinal goblet cells[8].

Tuft cells are a rare population of chemo-sensory epithelial cells that line the gastrointestinal and respiratory tracts[9,10], and are

[1]Olivia Newton-John Cancer Research Institute, Heidelberg, Australia. [2]School of Cancer Medicine, La Trobe University, Bundoora, Australia. [3]Walter and Eliza Hall Institute of Medical Research, Melbourne, Australia. [4]Department of Medical Biology, University of Melbourne, Melbourne, Australia. [5]Department of Medicine, University of California San Francisco, San Francisco, USA. [6]Howard Hughes Medical Institute, University of California San Francisco, San Francisco, USA. [7]These authors jointly supervised this work: Matthias Ernst, Michael Buchert. ✉e-mail: Michael.buchert@onjcri.org.au

identified by expression of Doublecortin-like kinase 1 (DCLK1) in mice, or Choline acetyltransferase (ChAT) in humans, respectively[11]. Currently, the role tuft cells play in the epithelium is uncertain, with tuft cells previously being identified as a quiescent stem cell[9], other studies have found tuft cells to rarely proliferate or display stem cell characteristics[12,13]. In a homeostatic setting DCLK1 identifies post mitotic gastrointestinal tuft cells[14]. While during chemically or genetically inducible tumor development, DCLK1[+] cells are proposed to be tumor stem cells and reserve stem cells[15,16], with long-lived DCLK1[+] tuft cells reported to act as cancer-initiating cells in the colon and intestine[9,14–17]. In addition, in response to helminth infections and other external stimuli, tuft cells secrete cytokines (e.g. interleukin (IL) 25), inflammatory mediators (e.g. eicosanoids), neurotransmitters (e.g. acetylcholine), and other signaling molecules to promote immune cell activation and restore tissue homeostasis[10,18–24]. Elevated IL25 has been associated with inflammatory bowel disease, as well as an increase in epithelial production of IL33, IL6, and TNFα[25]. While decreased IL25 was observed in inflamed mucosal tissue of IBD patients[26], and IL25 deficiency in mice conferring resistance dextran sulfate sodium-induced colitis[25]. Emerging evidence suggests that tuft cells may also orchestrate early oncogenic processes, as suggested by their rapid expansion and cancer stem cell-like properties observed in pre-neoplastic lesions of the gastrointestinal tract[9,27,28]. Within gastro-intestinal tissues, tuft cell-derived IL25 promotes the activation of type 2 innate lymphoid cells (ILC2s), and their subsequent production of IL13 stimulates the expansion of tuft cells[20–22].

Recent findings distinguish between ILC2s in resting and effector states[29,30], which broadly correspond to natural (nILC2s) and inflammatory (iILC2s) cells[31]. Tissue-resident nILC2s are characterized by the expression of the IL33 receptor (ST2) and are involved in maintaining epithelial barrier homeostasis and repair[32–34]. In contrast, iILC2s lack ST2 receptor expression, and are recruited into mucosal tissues where they expand in situ in response to infection-associated IL25 signaling via their IL17RB receptor subunit[32,35,36]. Although ILC2s are best understood for their contribution to immune defense against intestinal parasites, they are increasingly recognized as a novel immune cell type regulating anti-tumor immune responses[37–41]. Moreover, IL33 responsive ILC2s have been linked to *Helicobacter pylori* driven gastric metaplasia in humans and mice[37]. In addition, IL33 activated ILC2s were proposed as a source of IL13 during chemically induced metaplasia[42,43], with depletion of ILC2s resulting in reduced tuft cell hyperplasia and gastric metaplasia[43]. While ILC2s were found to be increased in the blood of gastric cancer patients[44], little else is known about their interactions with gastric tumor development. Tuft cells and ILC2s have been implicated as a driver for epithelial stem cell proliferation and tissue remodeling in the small intestine[45,46], however it remains unclear whether these cells contribute to the initiation and progression of GC. Here, we provide complementing evidence that a cytokine-supported tuft cell-ILC2 circuit, optimized to combat intestinal helminth infection, becomes coerced to underpin gastric metaplasia and cancer in mice, and remains evident as a therapeutic vulnerability in human GC.

## Results

### Tuft cells and ILC2s are increased during Spasmolytic polypeptide-expressing metaplasia (SPEM), a precursor to gastric cancer

Metaplasia is the leading risk factor for GC, while increased abundance of tuft cells and ILC2s in the gastric mucosa has been associated with *H. pylori* infection[47,48] and metaplasia[43,49–51]. We therefore induced gastric metaplasia through administration of high dose tamoxifen (HDTmx; 250 mg/kg)[52], which has been reported to induce a similar phenotype to that seen as a result of chronic *Helicobacter* infection[52–54]. Following treatment with HDTmx we observed the characteristic expansion of Gastric intrinsic factor (GIF)/GSII-lectin metaplastic cells, with

concomitant trans-differentiation of gastric chief cells into a spasmo-lytic polypeptide-expressing metaplastic (SPEM) cell type represented by TFF2 expression (Supplementary Fig. 1a, b)[52]. Two days after HDTmx administration, the metaplastic transformation was characterized by an increased abundance of SiglecF[+]CD24[+]EpCAM[+] tuft cells and KLRG1[+]CD90.2[+] ILC2s when compared to vehicle-treated controls (Supplementary Fig. 1c, d). The increase in ILC2s was primarily attributed to the IL25-responsive iILC2 subpopulation (Supplementary Fig. 1e), rather than the IL33-responsive nILC2s cells (Supplementary Fig. 1f), suggesting that SPEM expands iILC2s independently of nILC2s. In addition, we observed increased epithelial proliferation and cell death alongside the development of SPEM (Supplementary Fig. 2a).

To clarify the functional contribution of tuft cells to metaplasia, we generated *BAC(Dclk1::CreERT2);Rosa26^{DTA/+}* mice to enable diphtheria toxin A (*DTA*)-dependent ablation in DCLK1-expressing tuft cells[55]. Accordingly, we obtained tuft cell-depleted (TC[Δ]) mice following induction of Cre recombinase activity in *BAC(Dclk1::CreERT2);Rosa26^{DTA/+}* mice either in response to low dose tamoxifen (LDTmx; 50 mg/kg,), or with HDTmx to simultaneously induce SPEM in the resulting TC[Δ] mice (Fig. 1a). We observed partial protection from SPEM in mice of the HDTmx TC[Δ] cohort that occurred in wild-type tuft cell proficient CreERT2-negative *Rosa26^{DTA}* (TC[WT]) mice (Fig. 1b). Indeed, while the TC[Δ] mice from the HDTmx and LDTmx-treatment cohorts showed a more than 90% reduced abundance of gastric tuft cells (Fig. 1c), only the HDTmx TC[Δ] cohort had a reduced presence of ILC2s (Fig. 1d). This difference was primarily accounted for by a reduction of the more abundant IL25-responsive iILC2s, rather than IL33-responsive nILC2s (Fig. 1e, f). Moreover, whilst the HDTmx-induced SPEM resulted in increased proliferation (Ki67) and cell death (Cleaved caspase 3) in the gastric mucosa compared to vehicle-treated TC[WT] mice (Supplementary Fig. 2a), in HDTmx-treated TC[Δ] mice there was reduced proliferation and increased cell death when compared to HDTmx-treated TC[WT] mice (Supplementary Fig. 2b). Collectively, our results suggest that the loss of tuft cells and/ or the associated reduction in ILC2s provides partial protection against the development of SPEM, while established SPEM is associated with a selective, and tuft cell-dependent increase of iILC2 in the gastric mucosa.

### Tuft cells and ILC2s are increased during gastric tumor development

Given the strong association between cancer development and gastric metaplasia, we next explored the role of tuft cells and ILC2s during early stages of tumor development. To do this we utilised the *gp130^{Y757F/ Y757F}* (*gp130^{F/F}*) mouse model, which spontaneously develop SPEM-associated intestinal-type gastric adenomas from 4 weeks of age (Supplementary Fig. 3a, b)[56]. These tumors develop due to the excessive activation of Stat3 in response to a tyrosine (Y) to phenylalanine (F) knock-in substitution mutation in the common IL6 family receptor gp130, preventing the binding of the negative regulator Suppressor of cytokine signaling (Socs3)[56].

Indeed, when compared to the mucosa of wild-type *gp130^{+/+}* mice, we detected increased proportions of DCLK1[+] and SiglecF[+]CD24[+]EpCAM[+] tuft cells in the adenomas of *gp130^{F/F}* mice (Supplementary Fig. 3c, d), which coincided with an increased abundance of iILC2s, but not nILC2s (Supplementary Fig. 3e-g). We therefore explored whether tuft cells contributed to gastric adenoma formation by generating *gp130^{F/F}BAC(Dclk1::CreERT2);Rosa26^{DTA/+}* (*gp130^{F/F}TC[Δ]*) compound mutant mice to enable inducible tuft cell ablation in response to LDTmx. We observed smaller tumors in *gp130^{F/F}TC[Δ]* mice compared to LDTmx-treated *gp130^{F/F}Rosa26^{DTA/+}* (*gp130^{F/F}TC[WT]*) mice that lack the Cre transgene (Fig. 2a–c). In addition, ILC2-deficient *gp130^{F/F}R5-IL5;^{dtTomato-IRES-Cre}Rosa26^{DTA/+}* (*gp130^{F/F}ILC2[Δ]*) compound mutant mice had a reduced tumor burdens compared to ILC2-proficient *gp130^{F/F}ILC2[WT]* mice (Fig. 2d). Consistent with the

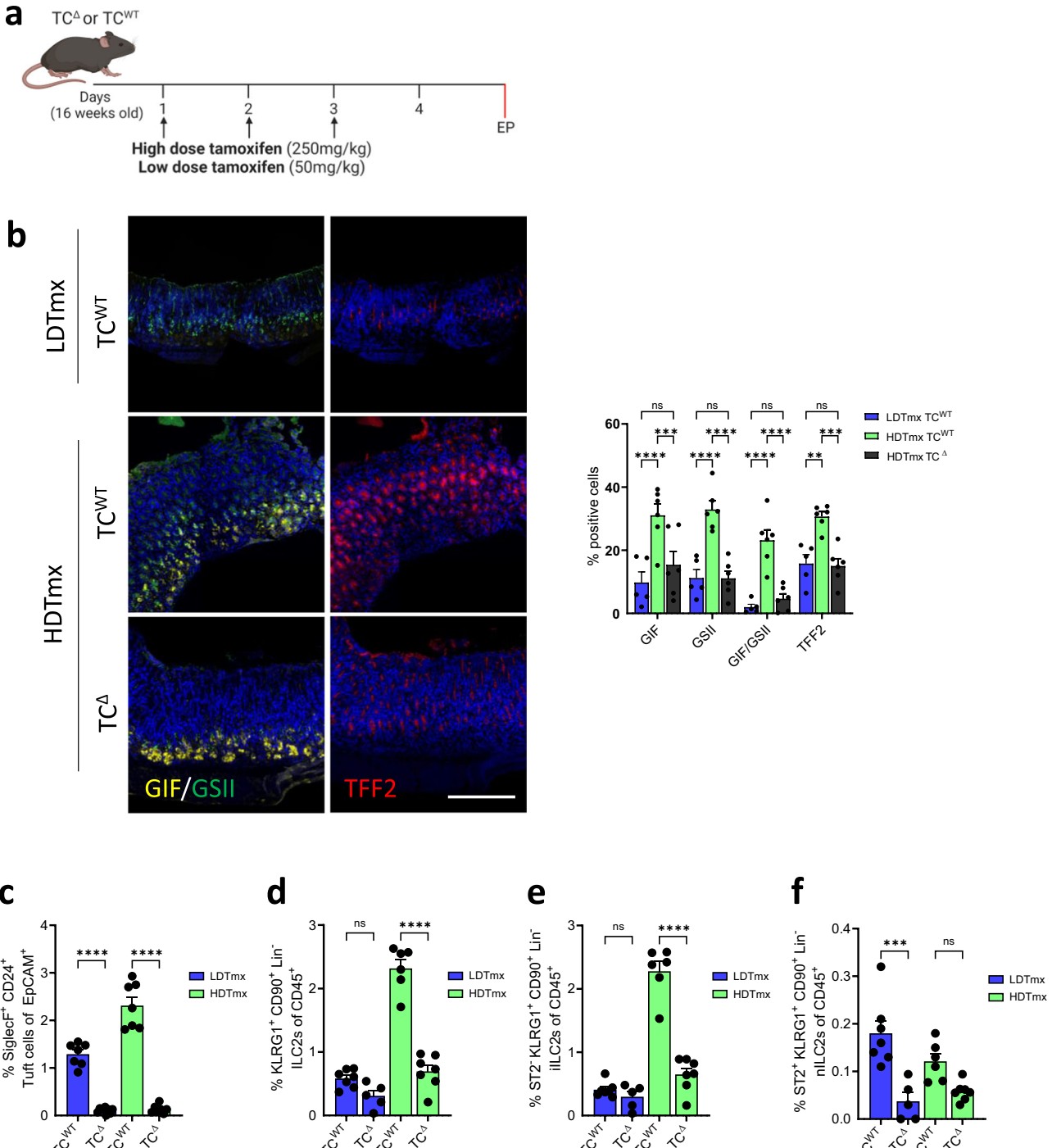

**Fig. 1 | Ablation of tuft cells results in reduced ILC2s in both WT and HDTmx treated mice. a** Schematic for experimental SPEM induction and tuft cell ablation. 16-week-old *BAC(Dclk1::CreERT2); Rosa26^DTA/+* (TC^Δ) or CreERT2-negative *Rosa26^DTA* (TC^WT) mice were treated with either HDTmx (250 mg/kg) once daily for 3 consecutive days to induce gastric spasmolytic polypeptide-expressing metaplasia (SPEM) and tuft cell ablation (in TC^Δ mice), or LDTmx (50 mg/kg) once daily for 3 consecutive days to induce tuft cell ablation (in *BAC(Dclk1::CreERT2);Rosa26^DTA/+* mice). EP = endpoint. Created with BioRender.com. **b** Representative Immuno-fluorescence staining and quantification of stomachs from TC^WT and TC^Δ mice following treatments as described in Fig. 1a and enumerated for Gastric intrinsic factor (GIF)/GSII-lectin positive and TFF2-positive SPEM cells. *N* = 5, 6 and 6 respectively. Scale bar = 200μm. (**c**) Flow-cytometry quantification of SiglecF+CD24+EpCAM+ tuft cells from stomachs of TC^WT and TC^Δ mice following

treatment with LDTmx or HDTmx. *N* = 7, 9, 7 and 7 respectively. **d** Flow-cytometry quantification of KLRG1+CD90.2+Lineage⁻CD45+ ILC2s in stomachs of TC^WT and TC^Δ mice following treatment with LDTmx or HDTmx. *N* = 7, 5, 6 and 7 respectively. **e** Flow-cytometry quantification of ST2⁻KLRG1+CD90.2+Lineage⁻CD45+ ILC2s in stomachs of TC^WT and TC^Δ mice following treatment with LDTmx or HDTmx. *N* = 7, 5, 6 and 7 respectively. **f** Flow-cytometry quantification of ST2+KLRG1+CD90.2+Lineage⁻CD45+ ILC2s in stomachs of TC^WT and TC^Δ mice following treatment with LDTmx or HDTmx. *N* = 7, 5, 6 and 7 respectively. Data represents mean ± SEM, *p* values from one-way ANOVA and Tukey's multiple comparisons tests **$p < 0.01$, ***$p < 0.001$, ****$p < 0.0001$, ns - not significant. Each symbol represents an individual mouse. Data is from two pooled experiments. Source data and exact *p* values are provided as a Source Data file.

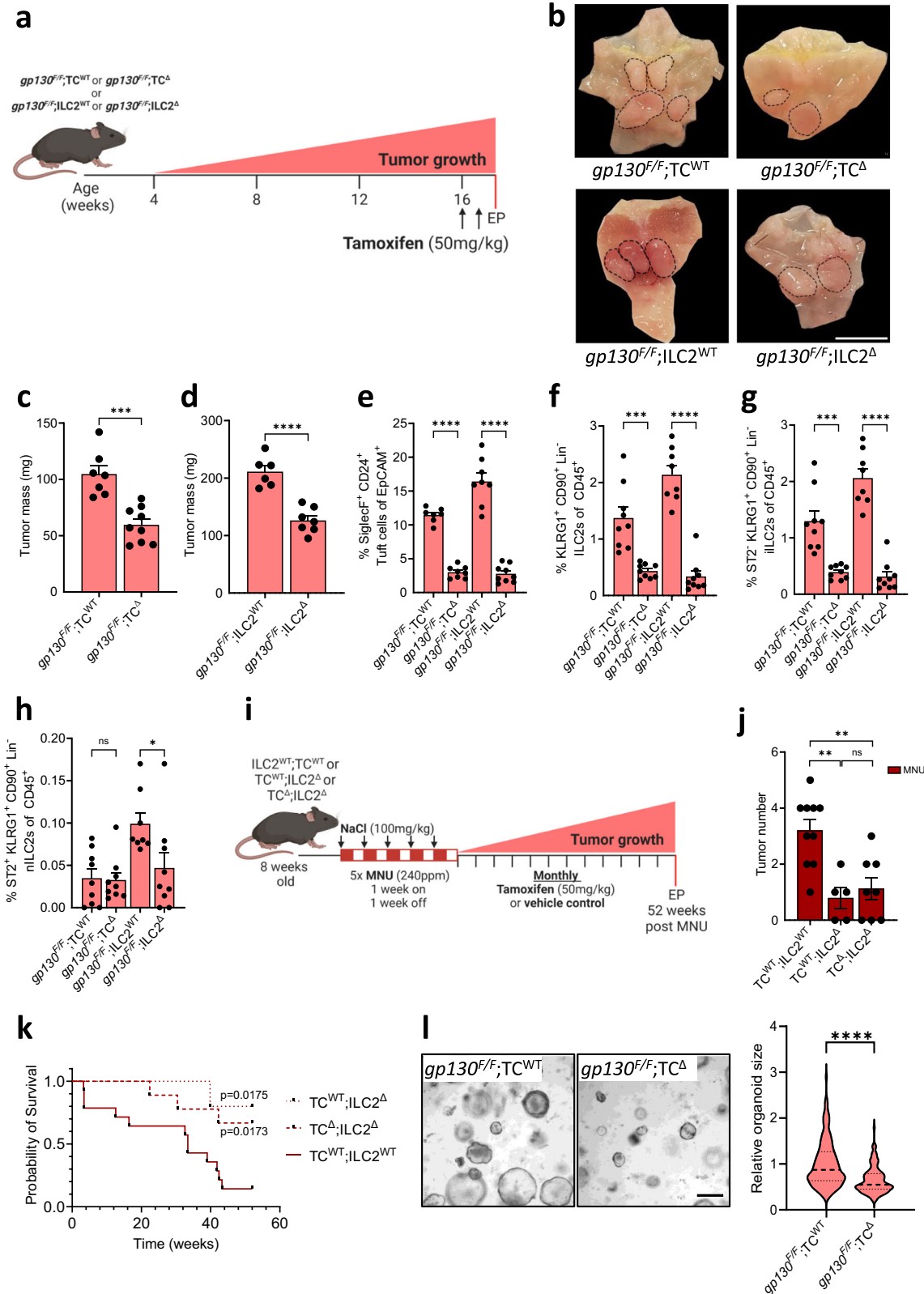

functional relationship between tuft cells and ILC2s observed during SPEM, we detected significantly fewer tuft cells in $gp130;^{F/F}ILC2^{\Delta}$ mice compared to $gp130;^{F/F}ILC2^{WT}$ mice (Fig. 2e). Conversely, we observed a significant reduction in total ILC2s within the stomachs of $gp130;^{F/F}TC^{\Delta}$ mice (Fig. 2f), which was attributed primarily to a reduced abundance of iILC2s (Fig. 2g, h). Finally, tumors from $gp130;^{F/F}ILC2^{\Delta}$ and $gp130;^{F/F}TC^{\Delta}$ mice showed reduced epithelial proliferation and increased cell

death compared to tumors recovered from the corresponding LDTmx-treated $gp130;^{F/F}TC^{WT}$ and $gp130;^{F/F}ILC2^{WT}$ mice (Supplementary Fig. 4a, b). Collectively, this data provides a compelling argument that tuft cells and ILC2s directly promote the development of tumors through the same mechanism by which these cell types promote SPEM.

To explore the contribution of tuft cells and ILC2s to the formation of SPEM-independent sporadic gastric cancer, we utilised the N-

**Fig. 2 | Genetic ablation of tuft cell reduces gastric adenoma growth.**
**a** Schematic outline of the genetic mouse models used to ablate tuft cells and ILC2s. Tuft cell ablation was achieved in *gp130;F/F*TC$^\Delta$ mice and compared against *gp130;F/F*TC$^{WT}$ littermate controls. Constitutive ILC2 ablation in *gp130;F/F*ILC2$^\Delta$ mice were compared against *gp130;F/F*ILC2$^{WT}$ littermate controls. All cohorts were given two injections of tamoxifen to induce tuft cell ablation at 16-weeks of age. EP = endpoint. **b** Representative wholemounts of stomachs of *gp130F/F* compound mutant mice as described in Fig. 2a. Black dotted circles indicate tumors. Scale bar = 8 mm. **c** Tumor mass of 17-week-old compound-mutant *gp130F/F* mice following genetic tuft cell ablation (*gp130;F/F*TC$^\Delta$) compared to Tmx-treated *gp130;F/F*TC$^{WT}$. N = 7 and 9 respectively. **d** Tumor mass of 17-week-old ILC2$^\Delta$ compared to age matched ILC2$^{WT}$. N = 6 and 7 respectively. **e** Flow-cytometry quantification of SiglecF$^+$ CD24$^+$ EpCAM$^+$ tuft cells in tumors of the indicated genotypes treated as described in Fig. 2a. N = 7, 8, 8 and 9 respectively. **f** Flow-cytometry quantification of ILC2s as KLRG1$^+$CD90.2$^+$Lineage$^-$CD45$^+$ in tumors of the indicated genotypes treated as described in Fig. 2a. N = 9, 9, 8 and 9 respectively. **g**, **h** Flow-cytometry quantification of iILC2s and nILC2s in tumors of the indicated genotypes treated as

described in Fig. 2a. N = 9, 9, 8 and 9 respectively. **i** Schematic outline of the MNU/NaCl-induced GC mouse model and experimental mouse strains. TC;$^\Delta$ILC2$^\Delta$, TC;$^{WT}$ILC2$^\Delta$, and ILC2;$^{WT}$TC$^{WT}$ mice were treated with MNU for 5 alternating weeks, then were aged for 52 weeks with monthly tamoxifen injections to ablate tuft cells. EP = endpoint. Created with BioRender.com. **j** Quantification of tumor numbers in MNU/NaCl-treated mice as described in Fig. 2i. N = 10, 5 and 8 respectively. **k** Kaplan−Meier survival analysis of MNU/NaCl-treated mice, TC;$^{WT}$ILC2$^\Delta$ *p = 0.0175 (cp to TC;$^{WT}$ ILC2$^{WT}$), TC;$^\Delta$ILC2$^\Delta$ *p = 0.0173 (cp to TC;$^{WT}$ILC2$^{WT}$, Mantel-Cox test). N = 14, 5 and 9 respectively. **l** Representative images and quantification of relative organoid diameter of 10-day old organoids derived from, *gp130;F/F*TC$^{WT}$ and *gp130;F/F*TC$^\Delta$ mice. Scale bar = 300μm. n = 288 and 116 respectively. Data represents mean ± SEM, p values from two-sided Student's t-test or one-way ANOVA and Tukey's multiple comparisons tests *p < 0.05, **p < 0.01, ***p < 0.001, ****p < 0.0001, ns - not significant. Each symbol represents an individual mouse. All Data is from two pooled experiments. Source data and exact p values are provided as a Source Data file.

Methyl-N-Nitrosourea (MNU) model[57], and administered this carcinogen to ILC2-deficient *R5-IL5;dtTomato-IRES-CreLSL-Rosa26DTA* mice (referred to as TC;$^{WT}$ILC2$^\Delta$ mice[58]) or compound mutant mice to achieve simultaneous ablation of ILC2 and tuft cell (TC;$^\Delta$ILC2$^\Delta$) (Fig. 2i). Compared to ILC2- and tuft cell-proficient TC;$^{WT}$ILC2$^{WT}$ mice, we observed fewer gastric adenocarcinomas in TC;$^{WT}$ILC2$^\Delta$ and TC;$^\Delta$ILC2$^\Delta$ mice (Fig. 2j), which correlated with their prolonged survival compared to carcinogen-challenged TC;$^{WT}$ILC2$^{WT}$ mice (Fig. 2k).

To unambiguously validate tuft cells as the population that promotes tumor growth, we established epithelial organoids from the gastric tumors of *gp130;F/F*TC$^\Delta$ mice. As a result we observed significantly reduced growth of tuft cell-deficient *gp130;F/F*TC$^\Delta$ organoids, when compared to their corresponding tuft cell-proficient *gp130;F/F*TC$^{WT}$ counterparts (Fig. 2l).

Collectively, we surmise from these observations that ILC2s serve as key enablers of gastric adenocarcinoma formation by directly regulating the abundance of tuft cells that laterally promote the growth of transformed gastric epithelium.

## Tuft cell - ILC2 signaling is primarily through IL13 and IL25

The increased abundance of tuft cells and ILC2s in the gastric mucosa and adenomas of *gp130F/F* mice is reminiscent of an intestinal anti-helminth immune response, where the detection of helminth metabolites by tuft cells results in the secretion of IL25 and associated expansion of intestinal iILC2s[23,45]. In turn, iILC2s secrete IL13 which promotes the differentiation of intestinal progenitor cells towards goblet cell and tuft cell lineages[20-22]. To confirm the existence of a similar mechanism in response to SPEM, we treated wild-type *gp130+/+* mice with HDTmx and measured the expression levels of the tuft cell and ILC2 markers *Dclk1* and *Gata3*, alongside their corresponding cytokines *Il25* and *Il13*, finding that all are increase during SPEM (Supplementary Fig. 5a). We then confirmed the role of tuft cells in maintaining this circuit during SPEM, finding that *Gata3* and *Il13* expression was impaired in HDTmx-treated TC$^\Delta$ mice (Supplementary Fig. 5b). Likewise, we observed elevated expression of *Dclk1, Il25, Gata3* and *Il13* in tumors collected from either *gp130F/F* mice, or from MNU-treated mice (Supplementary Fig. 5c−d). This was reversed in tumors recovered from the corresponding tuft cell-deficient *gp130;F/F*TC$^\Delta$ mice as well as from ILC2-deficient *gp130;F/F*ILC2$^\Delta$ mice (Supplementary Fig. 5e−f).

To demonstrate the responsiveness of tuft cells and ILC2s to IL13 and IL25, respectively, we pooled and FACs-purified CD45.2$^+$ hemopoietic and EpCAM$^+$ epithelial cells isolated from stomachs of 10 wild-type *gp130+/+* and from tumors of 10 *gp130F/F* mice for single-cell transcriptomic analysis (Fig. 3a). tSNE-clustering classified 6.1% of all viable EpCAM$^+$ cells in *gp130F/F* mice as tuft cells compared to 4.3% in *gp130+/+* mice (Fig. 3b). Irrespective of genotype, tuft cells accounted

for the cell type with the highest expression of the *Il13ra1* gene (encoding the IL13 receptor) (Fig. 3c). Meanwhile, ILC2s accounted for 0.42% and 0.19% of all CD45.2$^+$ cells in tumors of *gp130F/F* and gastric mucosa of *gp130+/+* mice (Fig. 3b), and ILC2s were the dominant cell population to express the IL25 receptor gene *Il17rb* (Fig. 3c). As we were unable to detect IL13 expression across our single cell datasets, we FACs-purified tuft cells and ILC2s from the stomachs of *gp130+/+* and *gp130F/F* mice, as well as the remaining epithelial cells and CD45$^+$ immune cells. Analysing these populations we found that tuft cells expressed significantly more *Dclk1, Il25,* and *Il13Ra1* transcripts than their (tuft cell-depleted) EpCAM$^+$ epithelial counterparts (Fig. 3d). In addition, we observed that *Il13* was predominantly expressed in *gp130F/F* ILC2s, while expression of *Gata3* and *Il17rb* was increased in ILC2s compared to (ILC2-depleted) CD45$^+$ cells isolated from *gp130+/+* and *gp130F/F* mice (Fig. 3e). Together this data leads us to conclude that gastric tuft cells and ILC2s, like their intestinal counterparts, are the primary source of IL25 and IL13 during SPEM and gastric tumor development.

## Pharmacologic inhibition of the tuft cell - ILC2 circuit reduces gastric tumor growth

Due to the established link between IL25 and IL13, as well as the functional requirement we established for tuft cells and ILC2s during gastric metaplasia and tumor growth (Figs. 1−3), we next investigated the extent by which these cytokines promote epithelial growth. Consistent with the expression of *Il13ra1* and *Il17rb* on tuft cells, IL25 or IL13 stimulation of tumor organoids increased their size, while treatment of organoids with an α-IL25 neutralizing antibody resulted in reduced organoid growth (Fig. 4a). To explore whether these in vitro dependencies of gastric tumor organoids could translate to therapeutic approaches in vivo, we treated tumor-bearing *gp130F/F* mice with α-IL13 or α-IL25 neutralizing antibodies (Fig. 5a). We observed smaller tumors in mice that had undergone either treatment (Fig. 5b−d). Consistent with a role for IL13 and IL25 in establishing a tumor promoting feed-forward circuit involving tuft cells and ILC2s, tumors from antibody-treated mice showed reduced proportions of tuft cells (Fig. 5e, f) and iILC2s (Fig. 5g, h), as well as diminished expression of the corresponding cell type-specific genes *Dclk1/Il25* and *Gata3/Il13* (Supplementary Fig. 5g, h). In addition, we observed reduced epithelial staining for Ki67 and increased presence of cleaved caspase 3 in these tumors when compared to tumors of IgG-treated *gp130F/F* mice (Supplementary Fig. 6a, b). From this data, we surmise that tuft cell-derived IL25, and ILC2-produced IL13 support gastric tumor progression through the establishment of a feed-forward circuit comprising epithelial tuft cell in the tumor and hematopoietic ILC2s in the tumor microenvironment to collectively sustains epithelial proliferation and survival of gastric tumor cells.

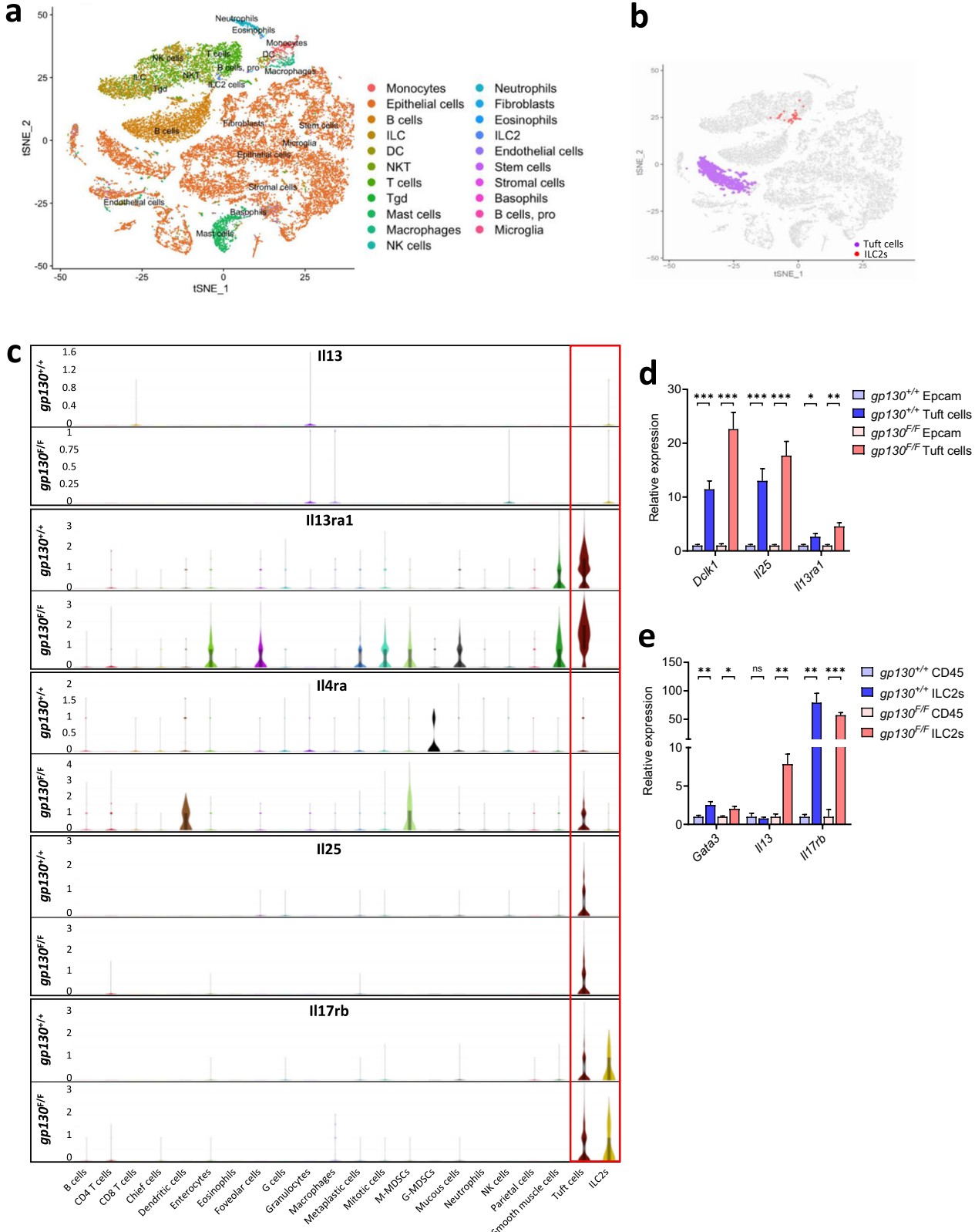

**Fig. 3 | Tuft cells and ILC2s promote tumor growth via IL25 and IL13 signaling.**
**a** t-SNE plot representing cell populations identified through unbiased cell clustering of pooled CD45.2+ and EpCAM+ cells sorted from gastric tissue and tumors of *gp130+/+* (*n* = 10) and *gp130F/F* (*n* = 10) mice. **b** Cell clusters identified as tuft cells (purple) or ILC2s (red) using SingleR against the ImmGene database. **c** Violin plots displaying the distribution and expression of *Il13, Il13ra1, Il4ra, Il25,* and *Il17rb* across immune and epithelial cell populations of *gp130+/+* and *gp130F/F* mice. **d** qRT-PCR analysis of *gp130+/+* and *gp130F/F* sorted EpCAM+ and tuft cells (SiglecF+CD24+) for the expression of tuft cell genes *Dclk1, Il25* and *Il13ra1* (*n* = 6 mice). **e** qRT-PCR analysis of *gp130+/+* and *gp130F/F* sorted CD45+ and ILC2s for the expression of *Gata3, Il13,* and *Il17rb* (*n* = 6 mice). Data represents mean ± SEM, *p* values from two-sided Student's *t*-test or one-way ANOVA and Tukey's multiple comparisons tests *\*p* < 0.05, *\*\*p* < 0.01, *\*\*\*p* < 0.001, ns - not significant. Source data and exact *p* values are provided as a Source Data file.

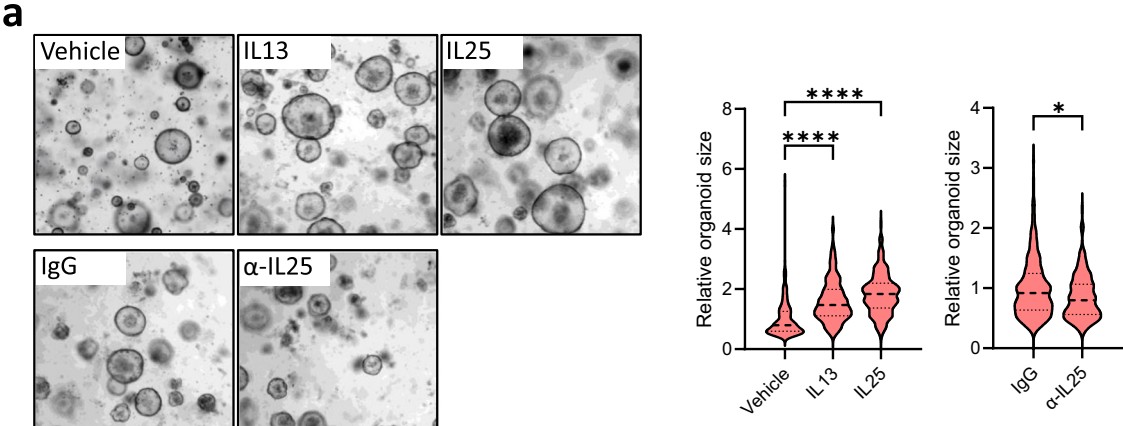

**Fig. 4 | Inhibiting tuft cell-driven IL25 signaling inhibits the formation and growth of gastric tumor organoids. a** Representative images and quantification of relative organoid diameter of 10-day-old organoids derived from *gp130^{F/F}* tumors and treated with either PBS (vehicle), IgG, IL13, IL25, or α-IL25 at 20 ng/ml for 7 days. Scale bar = 300 μm. Organoid *n* = 217 (vehicle), 325 (IL13), 322 (IL25), 293 (IgG) and 241 (α-IL25). Created with BioRender.com. Data represents mean ± SEM, *p* values from two-sided Student's *t*-test **p* < 0.05, ****p* < 0.0001. Data is from two pooled experiments, each comprising 4 domes of organoids per group. Source data and exact p values are provided as a Source Data file.

## Abundance of tuft cells and ILC2 correlate with GC patient survival

The involvement of tuft cells and ILC2s during SPEM and GC in mice, prompted us to investigate whether a gastric tuft cell-ILC2 signaling circuit is also present in humans and if it could predict clinical outcome in GC patients. As such, we interrogated survival data from both diffuse type and intestinal type gastric cancer of disease separately using our manually curated gene signatures for human tuft cell (*ChAT, IL25, POU2F3, TSLP, ALOX5, COX1*, and *AVIL*)[59], and ILC2s (*GATA3, IL13, ICOS, KLRG1, CRTH2, IL5*, and *IL4*). Indeed, we found significant correlation for both signatures and survival for the intestinal-type GC cohort (Fig. 6a). By contrast, we observed a much weaker correlation between our TC signature and survival in the diffuse-type GC cohort and the latter showed no correlation with our ILC2 signature (Fig. 6b). Given the association of SPEM with intestinal-type rather than the diffuse-type of GC in humans, and to substantiate the striking correlation between expression signatures and patient outcomes for intestinal-type GC, we used multiplex immunohistochemistry on tissue micro-arrays of intestinal-type GC to identify GATA3⁺CD3⁻ ILC2s and choline O-acetyltransferase (ChAT⁺) tuft cells, respectively[57,60]. This analysis revealed strong co-existence of ILC2 and tuft cells in 40% of specimen, compared to 20% or less of specimen showing expression of only one of these makers (Fig. 6c, Supplementary Fig. 7a). We therefore propose that the pro-tumorigenic tuft cell-ILC2 circuit in mice remains conserved in humans and is likely to functionally contribute to the progression of human intestinal-type GC (Fig. 6d), and by extension may provide therapeutic targets akin to those we identified in mice.

## Discussion

The susceptibility of cancer promotion to interference with anti-cytokine (signaling) therapy provides novel and exciting therapeutic opportunities to target cancer cell intrinsic hallmarks as well as shape the stromal and immune response of the tumor environment. Here, we describe the regulatory circuit between epithelial tuft cells and hematopoietic ILC2s connected through the reciprocal production of, and response to, IL25 and IL13 cytokines. In the context of a muta-genized epithelium, either by harboring the *gp130^{F/F}* mutation or being exposed to MNU, our data suggests a role for tuft cells to act in an epithelial gate keeper role by virtue of responding to IL13. The latter serves as mediator of the inflammatory response emanating from infiltrating iILC2 alongside granulocytic and NK effector cells, which therefore act as the cell population(s) enabling an anti-apoptotic and proliferative epithelial response. This mechanism is further supported by a tuft cell-based and IL25-dependent feed-forward mechanism, which not only promotes ILC2 abundance and function (e.g. *GATA3, IL13* expression), but also augments tuft cell function in an autocrine loop.

Although the cytokine responsiveness of ILC2s is contextual, IL33 and IL25 remain the major drivers of activation[43,61], with prominent roles for IL25 during skin allergies[62], pulmonary fibrosis[63] and helminth defense[64]. Meanwhile, IL33 activated ILC2s play a major role in driving SPEM development, through the recruitment and polarization of macrophages[43,50,51]. In addition, IL13 production has been linked to mast cells, B cells, T cells, and macrophages during genetically induced gastric metaplasia, with inhibition of IL13 reducing metaplasia development[65]. While these processes rely heavily on IL13-producing ILC2s, and a link between ILC2s and tuft cell abundance has been identified[43,51], a clear role for IL25 has yet to be determined. Tuft cell and ILC2 numbers increase during gastric colonization with *H. pylori*[43,50,51], and a regulatory tuft cells-ILC2 circuit had been proposed to drive the clearance of invading intestinal parasites in murine models[22,66,67]. On the other hand, IL33 signaling-based genetic or antibody-mediated impairment of ILC2s also protect against chemi-cally induced gastric metaplasia[43,51]. Here we provide evidence that the hard-wired IL13-IL25 loop, which underpins the required expansion of tuft cell and ILC2 populations to overcome parasite infections, is coerced as a phylogenetic conserved mechanism responsible to drive epithelial metaplasia in the stomach.

It remains unclear how IL25-secreting tuft cells initially expand in the gastric mucosa. While SPEM-associated gastrin release by endo-crine cells has been suggested, lineage tracing experiments have failed to identify the gastric stem cell responsible for producing tuft cells during SPEM[50]. Others have shown that cysteine leukotrienes secreted by tuft cells were required to complement full IL25-dependent ILC2 activation during intestinal helminth clearance[66], although our data suggests that blocking IL25 alone is sufficient to impair iILC2 expan-sion and to curb gastric tumor development. In the small intestine IL13 signals to epithelial stem and progenitor cells to promote hyper-plasia of IL25-expressing tuft cells, while we observed *Il13ra* expression in gastric enterocytes and Foveolar cells, which is consistent with *Il13ra* expression described for chief cells and metaplastic SPEM cells[42]. In addition, reduced organoid growth observed in α-IL25 treated cultures is consistent with the previous identification of tuft cells in gastric organoids[9], and highlights the possibility of tuft cell− derived IL25 as

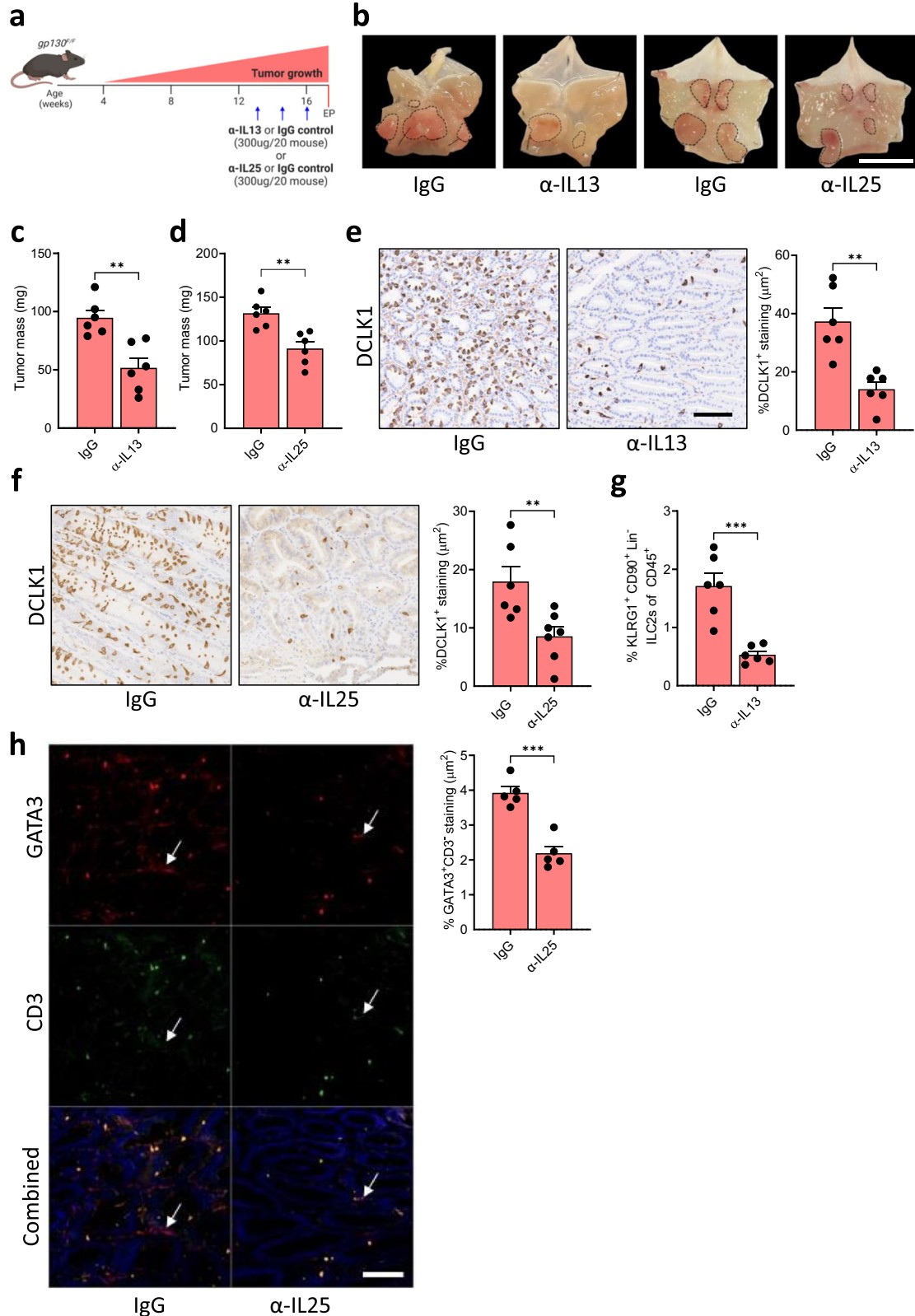

an autocrine acting growth factor given the relative absence of Il25 receptor expression in other epithelial cell types. While tuft cell−IL25 autocrine signaling has previously been mentioned in other publications, no functional impact of such autocrine signaling had been shown[21,68]. Given that we observe increased expression of *Il13ra1* in FACs-isolated tuft cells compared to other cells of the gastric epithelium in tumor of *gp130^F/F* mice, we speculate that tuft cell expansion

that occurs during metaplasia and adenoma formation, is driven by both inflammation-associated effectors cells as well as an autocrine/paracrine loop across the epithelium.

Based on our data and current literature, we propose a central role for ILC2 as an inflammatory sentinel producing IL13 not only in response primarily to IL25, but also to local accumulation of the alarmin IL33, consistent with findings that genetic ablation of either *Il33 or*

**Fig. 5 | Pharmacologic inhibition of the tuft cell-ILC2 circuit reduces gastric tumor growth. a** Schematic outline of the antibody-mediated blockade of IL13 or IL25. 13-week-old mice were treated with either α-IL13, α-IL25 or a matched IgG isotype control (300 μg/20 g mouse, once weekly for 3 weeks). Mice were culled one week after the third injection. EP = endpoint. **b** Representative images of *gp130^F/F* stomachs treated as described in Fig. 3a. Dotted circles indicate tumors. Scale bar = 8 mm. **c** Tumor mass of *gp130^F/F* mice following treatment with α-IL13 or a matched IgG isotype control. *N* = 6 and 6 respectively. **d** Tumor mass of *gp130^F/F* mice following treatment with α-IL25 or a matched IgG isotype control. *N* = 6 and 6 respectively. **e** IHC staining and quantification of DCLK1⁺ tuft cells in α-IL13 and IgG treated *gp130^F/F* mice. Scale bar = 300 μm. *N* = 6 and 6 respectively. **f** IHC staining

quantification of DCLK1⁺ tuft cells in α-IL25 and IgG treated *gp130^F/F* mice. Scale bar = 300 μm. *N* = 6 and 7 respectively. **g** Flow-cytometry quantification of KLRG1⁺CD90.2⁺Lineage⁻ CD45⁺ ILC2s in tumors of *gp130^F/F* mice treated with α-IL13 or a matched IgG isotype control. *N* = 6 and 6 respectively. **h** Immunofluorescence (IF) staining and quantification of Gata3⁺CD3⁻ ILC2s in tumors of *gp130^F/F* mice treated with α-IL25 or a matched IgG control. Arrows indicate ILC2s. Scale bar = 100 μm. *N* = 5 and 5 respectively. Data represents mean ± SEM, *p* values from two-sided Student's *t*-test *$p < 0.05$, **$p < 0.01$, ***$p < 0.001$. Each symbol represents an individual mouse. Data is from two pooled experiments. Source data and exact *p* values are provided as a Source Data file.

*Il13* expression prevented the formation of experimentally induced gastric metaplasia. In the latter case it is thought that IL33 is either released from damaged epithelium, or secreted by metaplasia-associated macrophages[42,43]. Thus, we surmise that during parietal cell loss, ILC2s are initially activated by either epithelial cell damage or macrophage-derived IL33, which in turn induces tuft cell expansion via IL13 signaling in gastric chief and/or metaplastic SPEM cells (Fig. 6d). Once the tuft cell population has sufficiently expanded, we show here that IL25 becomes the dominant driver for the expansion and activation of tissue-resident mucosal iILC2 at the expense of the IL33 responsive nILC2s which do not expand under these conditions. An additional and complementary role for IL25-responsive ILC2 has recently been identified in the APC^1322T/+ mouse model of intestinal polyposis, where iILC2-dependent activation of immune suppressive myeloid-derived suppressor cells (MDSC) served as a promoter of intestinal adenomas, resulting in impaired survival[69]. In addition, T cells have previously been shown to work with ILC2s in the clearance of helminth infections, where mice that lacked IL13 producing T cells had a reduced ILC2 response and helminth clearance[70]. Because of the limited antigenicity of the gastric tumors in *gp130^F/F* mice and our inability to detect IL13- or IL25-responsive MDSCs in these tumors, we cannot exclude an indirect contribution of IL25 signaling via MDSCs or IL13 produced by T cells to GC development.

Our study demonstrates that the tuft cell-ILC2 feed-forward circuit, originally identified as a repair mechanism of the intestinal epithelium during helminth infection, provides another facet of a wound-healing mechanism being hijacked to promote progression of neoplastic transformed cells. This occurs at the early metaplastic, adenomatous and later carcinoma stages and includes cytokine-dependent regulatory circuits that couple with local arising inflammatory triggers with an ensuing epithelial response. Intriguingly, the ILC2-tuft cell circuit is maintained by complementary IL25 and IL13 signaling between the two cell types arranged as non-redundant "single-point of failure" mechanisms. Accordingly, genetic interference of the circuit through ablation of either tuft cells or ILC2s, or therapeutic suppression of IL13 or IL25 signaling, confers profound therapeutic benefits at both earliest stages (i.e. gastric metaplasia) as well as later stages (i.e. gastric adenomas and adenocarcinomas) along the tumor trajectory. We predict that the functional insights from our preclinical models will be relevant to human GC, as tuft cell and ILC2 expression signatures were associated with poorer survival in patients with intestinal-type GC. A swift clinical translation of our discovery is supported by the availability of α-IL13 monoclonal antibodies that are currently optimized for the treatment of severe asthma[71] and the prospect of developing companion diagnostics for early detection of GC and patient stratification.

## Methods

### Study approval
All animal studies were conducted in accordance with the relevant ethical regulations for animal testing and research including the Australian code for the care and use of animals for scientific purposes. All animal studies were approved by the Animal Ethics Committee of Austin Health (A2019_05602, A2015_05289) or La Trobe University

(AEC 17-73). We have complied with all relevant ethical regulations for work with human participants. Usage of human gastric cancer tissues was approved by the Austin Health ethics committee (HREC/15/Austin/359) and informed consent was granted by all patients involved.

### Animal models
All mice were bred and maintained under specific pathogen-free conditions in the bioresource facilities of the La Trobe University or Austin Health. All strains were maintained on a 12-h light/dark cycle at constant temperature. Co-housed, age- and gender-matched littermates were utilized for all experiments. All interventions were performed during the light cycle on both male and female mice. All animals had free access to water and food (standard chow). The inducible *BAC(Dclk1::CreERT2)* strain has been previously reported[55] and was crossed with *LSL-Rosa26^DTA* model[72] to generate a mouse model of tuft cell ablation and with *gp130^Y757F* (*gp130^F/F*), a murine model of gastric cancer[56], to generate *gp130^F/F BAC(Dclk1::CreERT2);Rosa26^DTA/+*, a murine model of gastric cancer with inducible tuft cell ablation. The *R5-IL5^dtTomato-IRESCre LSLRosa26^DTA* model of constitutive ILC2 depletion has previously been reported[58] and was crossed with *gp130^F/F* or *BAC(Dclk1::CreERT2)* mice to generate the *gp130^F/F R5-IL5^dtTomato-IRESCre LSLRosa26^DTA* strain, a murine model of gastric cancer with constitutive ILC2 depletion, or *BAC(Dclk1::CreERT2);R5-IL5^dtTomato-IRESCre LSLRosa26^DTA* mice, respectively, a mouse model lacking ILC2s with optional (Tmx-inducible) tuft cell ablation. The control cohorts for tuft cell ablated mice, were comprised of CreERT2-negative *Rosa26^DTA*, or CreERT2-negative *gp130^F/F Rosa26^DTA* age matched littermates. Control cohorts of ILC2 depleted mice were comprised of *R5-IL5^+/+ LSLRosa26^DTA* or *gp130^F/F R5-IL5^+/+ LSLRosa26^DTA* mice age matched littermates.

### Tissue collection
Stomachs were removed, cut open along the greater curvature and flushed with cold PBS to remove contents. Stomachs were pinned out, with tumors being excised using curved scissors, taking care to avoid the mucosa. Tumors and stomach tissue were either fixed in 10% neutral buffered formalin (NBF) overnight at room temperature for histological analysis or snap frozen on dry ice and stored at −80 °C for molecular analysis.

### Tamoxifen treatment
Low dose tamoxifen (LDTmx) was prepared from tamoxifen (Sigma, T5648) at a concentration of 50 mg/ml in 10% ethanol and sterile sunflower oil. Mice were administered two doses of tamoxifen at 50 mg/kg via i.p. three days apart. Two days after the last tamoxifen dose mice were euthanized via CO2 asphyxiation and their stomachs were collected.

For the high dose tamoxifen (HDTmx) treatment, tamoxifen was prepared at a concentration of 250 mg/ml in 10% ethanol and sterile sunflower oil. Mice were administered tamoxifen at 250 mg/kg via i.p. once a day for three days. Two days after the last tamoxifen dose mice were euthanized via $CO_2$ asphyxiation and their stomachs were collected. Vehicle-treated control mice were administered 10% ethanol in sunflower oil.

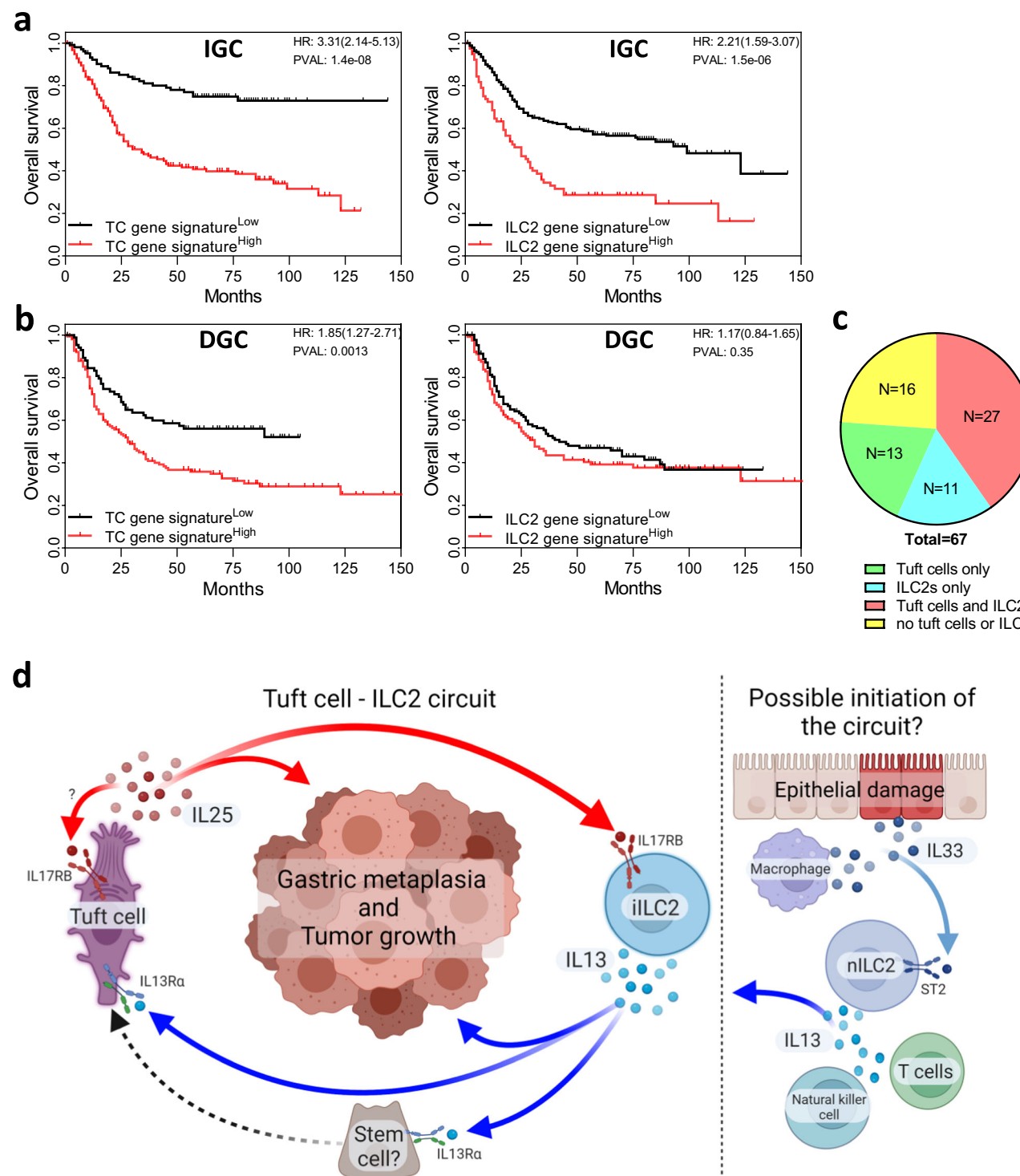

**Fig. 6 | Tuft cells and ILC2 are involved in human GC. a**, **b** Kaplan–Meier survival analysis for intestinal-type (IGC) and diffuse-type (DGC) GC patients segregated at the median level of gene expression for tuft cell (*ChAT, IL25, POU2F3, TSLP, ALOX5, COX1,* and *AVIL*) and ILC2 (*GATA3, IL13, ICOS, KLRG1, CRTH2, IL5,* and *IL4*) gene signatures. Data represents Kaplan–Meier plots depicting overall survival of GC patients from the GSE14210 (*n* = 145), GSE15459 (n-200), GSE22377 (n-43), GSE29272 (*n* = 268), GSE51105 (*n* = 94) and GSE62254 (*n* = 300) datasets. *P* value calculated with the Log-rank (Mantel-Cox) test. **c** Quantification of tuft cells and/or ILC2s in human intestinal-type GC tumor microarrays (Supplementary Fig. 8a) (*n* = 67 patients). **d** Tuft cell and ILC2 feed-forward circuit promotes gastric meta-plasia and tumor development through IL25 and IL13 signaling. Proposed initiation of the circuit through Natural killer cell, macrophage/epithelial or T cell produced IL33, leading to increased secretion of IL13 by activated nILC2s. Created with BioRender.com.

## MNU treatment

8-week-old mice were treated with a regimen of MNU (240 ppm) in the drinking water (1 week ON and 1 week OFF for 10 consecutive weeks). At the beginning of each MNU treatment week, mice were also administered 100 mg/kg of NaCl by oral gavage to increase the inci-dence of gastric tumor development[73]. 52 weeks after the last MNU treatment mice were euthanized via $CO_2$ asphyxiation and stomachs were collected and tumor numbers assessed.

## α-IL25 and α-IL13 treatment

13-week-old $gp130^{F/F}$ mice were given 1x weekly injection for 3 weeks of either α-IL25 (R&D Systems, MAB13992), α-IL13 (R&D Systems, MAB413) or IgG control (R&D Systems, MAB006 and MAB004) (at 300 μg/mouse). 1 week after the last injection, mice were euthanized via $CO_2$ asphyxiation. Stomachs were collected and tumors were excised and weighed before being fixed in 10% NBF overnight at room temperature.

## TCGA dataset analysis

Using the online KM plotter tool (https://www.kmplot.com/), we interrogated the patient survival against the median expression level of tuft cell related gene (*ChAT, AVIL, IL25*), and ILC2 related genes (*GATA3* and *IL13*) expression within following datasets; GSE14210, GSE15459, GSE22377, GSE29272, GSE51105, and GSE62254. We categorised intestinal-type gastric cancer patients and diffuse-type gastric cancer patients into groups with either high or low gene expression quartiles.

## Single cell RNA sequencing and analysis

For single cell capture and cDNA production, the 10X Genomics Chromium kit (v2) was used according to the 10x Single Cell 3' Protocol as previously described[74]. Single cell suspensions were prepared from the gastric tissues of 12-week-old $gp130^{+/+}$ and $gp130^{F/F}$ ($n = 10$ mice per genotype). Freshly sorted cells were pooled and manually counted before equal numbers per sample (1000 cells/μl) being loaded for capture. Sequencing was carried out on an Illumina Nextseq 500 with a maximum of 2 libraries per run. De-multiplexing, alignment to the mm10 transcriptome and unique molecular identifier (UMI)-collapsing to gene level against the NCBI RefSeq mouse (mm10) genome annotation build 38.1 inbuilt in Rsubread[75,76] were performed using cellCounts a function within Rsubread (v2.5.0)[77] for processing raw 10X scRNA-seq data. Cells with <200 genes detected or with high mitochondrial content of >40% were filtered out. Cells with >6000 genes detected were also filtered out to minimize the occurrence of doublets. Genes that did not map to official symbols were filtered as were genes that failed to express (an expressed gene has at least 1 UMI count) in at least 3 cells in at least 1 sample. After removal of unwanted cells from the dataset, the data was normalized using a global-scaling normalization method "LogNormalize" in Seurat[78]. Dimension reduction and cell clustering were performed using functions implemented in Seurat. An unbiased cell type annotation was performed using SingleR[79] against the ImmGen database which consists of normalized expression values of 830 microarray samples from pure populations of murine immune cells. To identify Tuft cells, we computed the relative activation of the known tuft cell marker genes in each cell as the average of the expression values for the known marker genes, a cell was annotated as a tuft-cell if its relative activation is >= the 95th percentile of all the relative expression values.

## RNA extraction and RT-PCR analysis

RNA extraction from whole tissue was performed using the RNeasy Mini Kit (QIAGEN, 74106), and cDNA was generated using the High-Capacity cDNA Reverse Transcription Kit (Applied Biosystems, 4368813) in accordance with the manufacturer's instructions. RNA extraction from sorted cells was performed using the RNA-easy Micro Plus kit (Qiagen), cDNA was generated with the SuperScript™ IV First-Strand Synthesis System (ThermoFisher) in accordance with the manufacturer's instructions. Quantitative RT-PCR analysis was performed using the SensiMix SYBR Hi-ROX Kit (Bioline, QT605-20) in duplicates (technical replicates) using the Viia7 Real-Time PCR System (Life Technologies). Samples were exposed to an initial denaturation step of 95 °C/10 min, followed by 40 cycles of amplification (95 °C for 15 s, 60 °C/1 min). *18S* or *Gapdh* were used as house-keeping genes, with fold changes in gene expression being calculated using the 2-ΔΔCT method. Primers used are outlined in Supplementary Table 1.

## Immunohistochemistry and quantification

Following fixation in 10% NFB, tissue was embedded in paraffin and cut into 10 mm thick sections. These sections underwent dewaxing and tissue hydration via incubation in xylene followed by gradient ethanol washes. Antigen retrieval was performed with citrate buffer heated in a microwave pressure cooker (pH 6 for 15 min), followed by blocking in 10% (v/v) normal goat serum for 1 h at room temperature. Primary antibodies as outlined in Supplementary Table 2, were diluted in 10% (v/v) normal goat serum and incubated overnight at 4 °C in a humidified chamber. Secondary HRP antibodies used were polyclonal rabbit anti-goat (Dako; P0449), polyclonal goat anti-rabbit (Dako; P0448), polyclonal goat anti-mouse (Dako; P0447), polyclonal goat anti-hamster (Abcam; ab6892). All antibodies were diluted in 10% (v/v) normal goat serum, incubated for 30 min at room temperature, and visualized using 3,3-Diaminobenzine (DAB, DAKO). Sections were counterstained with Mayer's hematoxylin for 10 s, developed in Scott's tap water for 20 s, then dehydrated in ethanol and xylene. Slides were cover slipped with mounting media and scanned using the Aperio ScanScope machine (ePathology). Quantification of stained sections was performed using ImageJ.

## Opal tissue staining on mouse tissue and human intestinal-type GC tumor microarrays

Opal staining was carried out using the Opal staining kit (akoyabio, OP7DS2001KT) and following the below protocol; 10 μm thick sections were dewaxed in xylene and tissue hydrated in gradient ethanol washes. Antigen retrieval was performed with citrate buffer heated in a microwave pressure cooker (pH 6 for 15 min), followed by blocking in 10% (v/v) normal goat serum for 1 h at room temperature. Primary antibodies were diluted in 10% (v/v) normal goat serum and incubated for 1 h at room temperature in a humidified chamber. Opal Polymer HRP was applied as a secondary antibody for 10 min at room temperature. The Opal fluorophore was then diluted in Amplification Diluent to a concentration of 1/50 and applied to the tissue for 10 min at room temperature. Sections were then stripped of all primary and secondary antibodies through antigen retrieval using citrate buffer heated in a microwave pressure cooker (pH 6 for 15 min) before the above process was repeated for each desired antibody. After the final antibody incubation, sections were incubated with spectral DAPI for 5 min at room temperature before mounting media was applied and sections were imaged using the Vectra imaging system. Tissue sections were then scanned using the Vectra and quantification of opal staining was performed using either InForm (Perkin Elmer) or Halo (Indica Labs).

## Preparation of single cell suspensions for flow cytometry

Flow cytometry was performed as previously described[80]. In short, tissues were cut into 1 mm pieces and digested in collagenase/dispase (Roche) and DNase I (Roche) in $Ca^{2+}$- and $Mg^{2+}$-free Hanks medium plus 5% FCS for 30 min at 37 °C with gentle shaking. Samples were then vortexed for 15 s, filtered and washed in PBS plus 5% FCS. Single cell suspensions were blocked with FC block (Invitrogen) for 20 min at 4 °C, before staining with fluorophore-conjugated primary antibodies (Supplementary Table 3) for 20 min at 4 °C in the dark. Cells were washed twice and re-suspended in PBS supplemented with 5% FCS prior to analysis with either an Aria III cell sorter or BD FACS Canto.

Isotype antibodies (Supplementary Table 3) and fluorescent-minus-one (FMO) controls were used to estimate background fluorescence in combination with either compensation beads and/or unstained controls. Dead cells were detected and excluded from analysis using Sytox Blue or Fixable Viability Dye, eF506. Tuft cells were identified as EpCAM+CD45$^{-/low}$CD24+SiglecF+ (Supplementary Fig. 8a). Inflammatory ILC2s were identified as ST2−KLRG1+CD90.2+Lineage− (CD11b−CD11c−CD19−Ly-6G−NK1.1−CD3−)

$CD45^+$. Natural ILCs were identified as $ST2^+KLRG1^+CD90.2^+$ $Lineage^-CD45^+$(Supplementary Fig. 8b). All experiments were analyzed with FlowJo software (Version 10).

## Organoid culture
Following the collection of gastric tissue, tissue was cut into small pieces and transferred to a 50 mL Falcon tube containing 20 mL of room temperature Gentle Cell Dissociation Reagent (Stem Cell technologies). The samples were then incubated at room temperature for 20 min on an orbital roller. Glands were released from the underlying tissue by shaking the tube vigorously for 20 seconds and supernatant was transferred to a new Falcon tube and centrifuged for 5 min at 1500 rpm (500 g) (4 °C). The supernatant was tipped off and the pellet was resuspended in 1 mL Advanced DMEM-F12 with Penicillin/Streptomycin (1/100) and passed through a 70 μm cell strainer. The number of glands in 10 μl was counted with a microscope, to determine the volume needed to have 100 glands per 50μl of Matrigel. Glands were centrifuged for 5 min at 1500 rpm (500 g) (4 °C) and supernatant was discarded. Glands were then resuspended in Matrigel, and 50 μl was pipetted into each well of a pre-warmed 24-well plate to create domes. 500 μl of IntestiCult™ Basal Medium (Stem Cell technologies) was added to each well, comprised of 90 ml IntestiCult™ Basal Medium, supplemented with 5 ml of the IntestiCult Supplement 1, 5 ml of Supplement 2 and Penicillin/Streptomycin (1/100). Organoids were grown at 37 °C, 10% $CO_2$ in a humidified incubator, changing the media every 3 days.

## Treatment of organoids
Two days after seeding into Matrigel, organoids were grown in the presence of either IL25 (R&D Systems, 1399-IL), IL13 (R&D Systems, 413-ML), α-IL25 (R&D Systems, MAB13992), α-IL13 (R&D Systems, MAB413) or either PBS or IgG control (R&D Systems, MAB006 and MAB004) antibodies (at 20 ng/ml). Media and antibodies were replaced every second day.

## Enzyme-linked immunosorbent assays (ELISA)
Were performed using the indicated R&D Systems DuoSet ELISA system (InVitro Technologies) on protein extracted from tissue. Samples were loaded in duplicate, and the protocol was carried out in accordance with the manufacturer's instructions, before carrying out the ELISA, samples were read by SPECTROstar Nano (BMG LABTECH). Protein concentration of the targets was then quantitated using the MARS Data Analysis Software (BMG LABTECH).

## Statistical analysis
All experiments were conducted at least twice with ≥3 sex- and aged-matched mice per group. Where drugs were administered, animals were randomized into their corresponding treatment groups. Tumor growth was measured and recorded by an independent assessor who was blinded to the experimental conditions. All measurements were taken from distinct samples. No data was excluded from the analysis. All groups were tested for normality with the Shapiro-Wilk test and unless stated were normally distributed. Comparisons between mean values were performed with a two-tailed Student's $t$-test for comparisons between two groups, or a one-way ANOVA and Tukey's multiple comparisons tests for comparisons between multiple groups using Prism 10 software (GraphPad). A $p$ value of <0.05 was considered statistically significant. All data is expressed as the mean ± SEM. Each 'n' or symbol represents a single mouse (biological replicate). The statistical Log-rank (Mantel-Cox) test, Hazard ratios (logrank) and median survival were calculated in Prism 10.

## Reporting summary
Further information on research design is available in the Nature Portfolio Reporting Summary linked to this article.

## Data availability
The Single cell sequencing data generated in this study have been deposited in the Gene Expression Omnibus database under accession code GSE217498. The publicly available data used in this study are available in the Kaplan–Meier Plotter database [https://kmplot.com/analysis/index.php?p=service&cancer=gastric] that uses the following datasets: GSE14210, GSE15459, GSE22377, GSE29272, GSE51105 and GSE62254. The remaining data are available within the Article, Supplementary Information or Source Data file. Source data are provided with this paper.

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

## Acknowledgements

We are indebted to David Williams (Department of Pathology, Austin Hospital) for providing the gastric cancer tissue arrays. We thank the members of the Cancer and Inflammation Program at the Olivia Newton-John Cancer Research Institute for helpful discussions and comments. Experimental schematics for Figs. 1a, 2a, i, 5a, 6d and Supplementary Fig. 1a were created with Biorender.com. Biorender publication licenses are provided and made out to our Cancer and Inflammation Program (CI Program). This research was possible dure to the following funding schemes and organisations, National Health and Medical Research Council of Australia (NHMRC) Principal Research Fellowship 1079257 (M.E.), NHMRC Program Grant 1092788 (M.E.), NHMRC Investigator Grant 1173814 (M.E.), NHMRC Project Grant 1143020 (M.B.), La Trobe RFA Understanding Disease Grant (M.B.), Operational Infrastructure Support Program, Victorian Government, Australia (M.B.), La Trobe University Graduate Research Scholarship (R.N.O.), NHMRC Peter Doherty Early Career Fellowship GNT1166447 (A.R.P.), NHMRC Project Grant 1143030 (C.S.), NHMRC Project Grant 1143030 (R.M.L.), National Institute of Health AI026918 (R.M.L.), Howard Hughes Medical Institute (R.M.L.), The Sandler Asthma Basic Research Center (R.M.L.), Cancer Council Victoria's Grant-in-Aid APP1160708 (M.F.E.) and Victorian Cancer Agency Mid-Career Research Fellowship MCRF20018 (M.F.E.).

## Author contributions

Conceptualization: M.B., M.E., R.N.O., and R.M.L. Methodology: R.N.O., M.B., R.M.L., B.P., C.S., D.C., W.S., and A.R.P. Investigation: R.N.O., M.B., A.L.C., D.C., W.S., S.A.S., and A.R.P. Resources: M.B., M.E., R.M.L., B.P., D.B., M.F.E., and W.S. Visualization: R.N.O., M.B., A.R.P., and D.C. Funding acquisition: M.B., R.M.L., C.S., and M.E. Project administration: M.B. and M.E. Supervision: M.B. and M.E. Writing—original draft: R.N.O. and M.B. Writing—review and editing: R.N.O., M.B., M.E., R.M.L., C.S., A.L.C., M.F.E., and A.R.P.

## Competing interests

The authors declare no competing interests.
