## [Peer Review File · Nature Communications]

A tuft cell - ILC2 signaling circuit provides therapeutic targets to inhibit gastric metaplasia and tumor developmentREVIEWER COMMENTS

Reviewer #1 (Remarks to the Author): with expertise in gastric cancer, tuft cells

Overall, this is a fascinating report that provides a substantial advance on previous reports demonstrating the link between intestinal tuft cells and an IL25 mediated ILC2 dependent immunomodulatory response. Indeed, this manuscript demonstrates the roles of gastric tuft cells on intestinal subtype gastric metaplasia leading to gastric carcinogenesis. These findings would appear to be incremental, however, the clear delineation linking SPEM progression to tuft cell abundance and likely activity opens a clear path to investigating the functional mechanism that drives gastric tuft cells toward a malignant phenotype. One potential major concern is the reliance on diphtheria toxin-mediated ablation of tuft cells which may clearly have other nonspecific effects on ILC2s however, the experimental controls provided in the supplemental figures and throughout the manuscript limit this concern. The manuscript is often difficult to read and figures are mislabeled or out of order.

Minor Issues:

Figure 1 legend. Panels are mislabeled. Panels A and B appear to be for the same Schematic and B does not match Figure 1B, which seems to be the description given for Panel C in the Legend. The legends D-G need to be labeled to match that of the Figure.

Figure 3B is not discussed till after the discussion of Figure 4.

Line 208 Superfluous “the” between “...the pro-tumorigenic the tuft cell...”.

Reviewer #2 (Remarks to the Author): with expertise in ILC2

The authors investigate the roles of IL-25-producing tuft cells and ILC2s in gastric metaplasia. Using a number of mouse models they demonstrate that reducing the numbers of tuft cells also reduces gastric metaplasia. Similarly, depleting IL-5-producing cells (which may be ILC2s) also reduces gastric metaplasia. Furthermore, anti-IL-25 and anti-IL-13 treatment reduces adenoma proliferation. The authors suggest that there is a feed forward loop in which in which IL-25 derived from tuft cells stimulates ILC2s to produce IL-13 which

in turn promotes gastric metaplasia and tumorigenesis. The authors also present human data. However, the human data plots appear to be mislabelled, making conclusions difficult.

This is an interesting area and the authors have performed a range of experiments which are suggestive of their proposed mechanism. However, the authors never convincingly prove that the relationships that they observe are causative. There are also issues with the quality of the data and the clarity of the manuscript.

Major comments:

Introduction

1. The introduction fails to adequately introduce the existing literature reporting the roles of ILC2s, IL-25 and IL-33 in cancer – or even gut/intestinal cancer.
2. The authors should also introduce existing literature relating to gastric/intestinal stem cell development and how disruption of this pathway can lead to tumorigenesis. This is important given the relationship of the cells and markers under study.

Results

3. In Fig S2 the authors outline the cell markers that they used to define tuft cells and ILC2s in the gastric tissues. It is surprising that the authors don't better define the tuft cell population e.g. defining protein express of DCLK1, IL-25R, IL-13R etc.
4. The definition of iILC2s and nILC2s raises concerns. In figure S2B the authors use an anti-IL-25R antibody to define iILC2s, however most of the experiments do not use this antibody to define these cells and instead cells are referred to as ST2-negative. It is not clear if these are ILC2s at all. Do the authors find that gastric ILC2s express IL-25R at homeostasis as reported elsewhere (PMID:32891625). The authors should look at single IL-25 and ST2 receptor-positive cells and IL-25R/ST2 double positive cells in their experiments. GATA3 staining should also be included. It is also not investigated whether these cells are local or recruited – to warrant being called iILC2s and nILC2s.
5. There are issues with the order of figures and parts of figures e.g. 3B is only cited after Figure 4 – this is not normal procedure.
6. The gp130^{Y757F}/Y757F mutant knock-in mouse needs to be introduced. Especially as many readers may assume F refers to Flox.
7. The authors need to perform flow cytometry in their models to define other immune cell

types. It is essential that CD4+ Th1 and Th2 cells are identified and CD8+ T cells.

8. This requirement is true also where the IL5-Tom-Cre mouse is used as the authors may also be depleting Th2 cells in the tissues or IL-5-producing myeloid cells.

9. In most experiments there is no mention of the number of experimental repeats to validate that the results are robust.

10. In Figure 3A/3B there is no indication of the numbers counted.

11. Fig 4B. The authors quotes percentages, but isn't this just a single data point when comparing WT with Tg (or are there more repeats?)?

12. Fig 4C- stain for IL-13R and IL-25R protein. These are tuft-like cells, but are induced by metaplasia, how similar are they to those in the small intestine?

13. In many experiments and figure legends it is unclear if experimental repeats have been performed (how many repeats and how many mice per group). These are essential to confirm the robustness of the data and author claims.

14. Only 3 data points in Fig 5E and no repeat.

15. What are all the GATA3-positive T cells in the images? They greatly outnumber the ILC2. Are these Th2 cells? Are they IL-13 producers? Are they IL-5 producers? Are they deleted in the IL5DTA mice? If they make IL-13 then why isn't this IL-13 active on epithelial cells over a 4 month model?

16. Are the other T cells CD8 cytolytic cells?

17. Figure 6A – this is confusing. Are the lines mislabelled? If not then isn't this contradictory with the main hypothesis?

18. Line 215. The use of 'novel' is not justified given the existing number of examples where this axis has been identified.

19. The authors have to better cite and discuss the already reported role of IL-33 (Mayer et al 2020 PMID 32891625) as these have also been studied in SPEM models. There is significant overlap in the phenotypes. How do IL-25 and IL-33 work alongside one another? This relationship is of significant importance and needs more attention here.

20. Line 257 – adenocarcinoma models were also used in these studies. Not Apcmin?

21. The authors do not undertake therapeutic treatment in adenocarcinoma 'cancer' models, but in pre-cancerous adenoma models. Could the authors use H pylori or TFF1-KO models to test cancer progression? I appreciate that the MNU model takes a year, but the 'control' of cancer is of significant interest.

22. To confirm their hypothesis the authors need to better link the cytokine and cell responses e.g. IL-25R or IL-13R deletion from tuft cells, IL-13 deletion from ILC2 etc, or perhaps use cell transfer models.

Minor comments

Line 123 should read gp130^{f/f};ILC2^{WT}

Line 170 refers to select subsets, but only ILC2 are shown. Were there supposed to be other cell types?

The term therapeutic vulnerabilities is confusing.

Line 263. What do the authors mean by 'phylogenetically optimized'? What is the evidence for this?

Fig 4A labelling too small.

Fig5A – a-IL-25, change to anti or alpha as used elsewhere.

Line 187 Fig s6G/H mentioned after s7A/B

Reviewer #3 (Remarks to the Author): with expertise in gastric metaplasia/cancer, immunology

This study proposes that pro-tumorigenic properties caused by the crosstalk between intestinal tuft cells and type 2 innate lymphoid cells (ILC2) may be promoting gastric metaplasia/tumorigenesis. Strengths of study include the use of multiple mouse models to investigate the role of ILC2s and tuft cells in gastric metaplasia/adenomas. These include the use of tamoxifen induced metaplasia in control and tuft cell deficient mice, tumor development in GP130^{F/F} mutant mice crossed to tuft cell deficient as well as ILC2 deficient mice as well as cytokine neutralizing antibodies. The data clearly show that tumor mass is reduced in GP130^{F/F} when tuft cells and ILC2s are deficient, and when mice are treated with IL-13 or IL-25 neutralizing antibodies. These data support a role for tuft cells and ILC2s in regulating tumor growth in these models. These results are noteworthy.

However, there are also some perceived weaknesses in the study. One is the lack of distinction between SPEM, which refers to the presence of MUC6 and TFF2 expressing metaplastic cells that share features with mucous producing deep antral glands from the

antrum of the stomach, and intestinal metaplasia, which share features with MUC2/TFF3 intestinal type goblet cells. These are distinct types of gastric metaplasia and this study does not distinguish between the two (Figure 1). It is difficult to determine how metaplasia is defined and quantified in Figure 1. In this model metaplasia (SPEM) is usually measured by TFF2/GIF/MUC6 expressing cells at the base of glands because TFF2 is also expressed in other cell types (neck cells, pit cells). Much of the cytokine and cytokine receptor expression data are correlative (mRNA expression), and only one IL-13 receptor is examined (IL13ra but not IL4ra). Thus, the proposed Tuft cell (IL25) – ILC2 (IL13) circuit, which is the unique aspect of this study, is not clearly established. Other studies have demonstrated roles for ILC2s and IL-13 in tuft cell hyperplasia and gastric metaplasia (most were cited, not PMID: 34587523), so clearly establishing the Tuft cell (IL25) – ILC2 (IL13) circuit would distinguish this study from others. Perhaps demonstrating that IL25 production specifically by tuft cells or IL-13 production specifically by ILC2s enhance gastric metaplasia/tumorigenesis. Also, the human datasets tuft cell signatures and ILC2 signatures correlate with survival curves in patients with intestinal type gastric cancer, but it is not clear why lower signature scores (in red) have less overall survival. This needs some explanation. Finally, the organoid experiments are presented in Figure 3, but discussed after Figure 4 in the results. It is not clear what are the sources of IL-25 (are there tuft cells in these organoids) that reduce growth in anti-IL25 treated organoids, or which cell types are responding to IL-25 when it increases the size of organoids. Some discussion would be helpful.

We thank all the reviewers for their helpful and insightful comments, concerns and questions. Please find below our detailed responses to every issue raised by each reviewer for our manuscript “*A tuft cell - ILC2 signaling circuit provides therapeutic targets to inhibit gastric metaplasia and tumor development*”.

Response to specific comments raised by **Reviewer #1:**

Reviewer Summary: “Overall, this is a fascinating report that provides a substantial advance on previous reports demonstrating the link between intestinal tuft cells and an IL25 mediated ILC2 dependent immunomodulatory response. Indeed, this manuscript demonstrates the roles of gastric tuft cells on intestinal subtype gastric metaplasia leading to gastric carcinogenesis. These findings would appear to be incremental, however, the clear delineation linking SPEM progression to tuft cell abundance and likely activity opens a clear path to investigating the functional mechanism that drives gastric tuft cells toward a malignant phenotype. One potential major concern is the reliance on diphtheria toxin-mediated ablation of tuft cells which may clearly have other nonspecific effects on ILC2s however, the experimental controls provided in the supplemental figures and throughout the manuscript limit this concern.”

Major comment 1. The manuscript is often difficult to read and figures are mislabeled or out of order.

We have gone through and adjusted the text throughout the manuscript to fix errors and improve the readability. We have also updated the order of figures to ensure they are discussed in ascending order.

Minor comment 1. Figure 1 legend. Panels are mislabeled. Panels A and B appear to be for the same Schematic and B does not match Figure 1B, which seems to be the description given for Panel C in the Legend. The legends D-G need to be labeled to match that of the Figure.

We have fixed the figure legend of Figure 1 to correctly match the data displayed in the figure panels with the text.

Minor comment 2. Figure 3B is not discussed till after the discussion of Figure 4.

We have updated the order of all figures to ensure that they are discussed in ascending order.

Minor comment 3. Line 208 Superfluous “the” between “...the pro-tumorigenic the tuft cell...”.

We have removed the superfluous “the” on PDF page 6 line 226 (previously was line 208).

Response to specific comments raised by **Reviewer #2**:

Reviewer Summary: “The authors investigate the roles of IL-25-producing tuft cells and ILC2s in gastric metaplasia. Using a number of mouse models they demonstrate that reducing the numbers of tuft cells also reduces gastric metaplasia. Similarly, depleting IL-5-producing cells (which may be ILC2s) also reduces gastric metaplasia. Furthermore, anti-IL-25 and anti-IL-13 treatment reduces adenoma proliferation. The authors suggest that there is a feed forward loop in which IL-25 derived from tuft cells stimulates ILC2s to produce IL-13 which in turn promotes gastric metaplasia and tumorigenesis. The authors also present human data. However, the human data plots appear to be mislabelled, making conclusions difficult.

This is an interesting area and the authors have performed a range of experiments which are suggestive of their proposed mechanism. However, the authors never convincingly prove that the relationships that they observe are causative. There are also issues with the quality of the data and the clarity of the manuscript.”

Comment 1. The introduction fails to adequately introduce the existing literature reporting the roles of ILC2s, IL-25 and IL-33 in cancer – or even gut/intestinal cancer.

In our revised manuscript, we have added the below information on ILC2s and their known roles in gastric metaplasia and cancer. (PDF page 2, line 69 – 76)

“Although ILC2s are best understood for their contribution to immune defence against intestinal parasites, they are increasingly recognized as a novel immune cell type regulating anti-tumor immune responses (1-5). Moreover, IL33 responding ILC2s have been linked to Helicobacter pylori driven gastric metaplasia in humans and mice (1). Additionally, IL33 activated ILC2s were proposed as a source of IL13 during chemically induced metaplasia (6, 7), with depletion of ILC2s resulting in reduced tuft cell hyperplasia and gastric metaplasia (7). While ILC2s were found to be increased in the blood of gastric cancer patients (8), little else is known about their interactions with gastric tumor development.”

Comment 2. The authors should also introduce existing literature relating to gastric/intestinal stem cell development and how disruption of this pathway can lead to tumorigenesis. This is important given the relationship of the cells and markers under study.

We have included the below additional information to explicitly refer to current literature outlining roles for tuft cells as epithelial stem cells. (PDF page 2, line 46 – 52)

“Currently, the role tuft cells play in the epithelium is uncertain, with tuft cells previously being identified as a quiescent stem cell (9), other studies have found tuft cells to rarely proliferate or display stem cell characteristics (10, 11). In a homeostatic setting DCLK1 identifies post mitotic gastrointestinal tuft cells (12). While during chemically or genetically inducible tumor development, DCLK1⁺ cells are proposed to be tumor stem cells and reserve stem cells (13, 14), with long-lived DCLK1⁺ tuft cells reported to act as cancer initiating cells in the colon and intestine (9, 12-15).”

Comment 3. In Fig S2 the authors outline the cell markers that they used to define tuft cells and ILC2s in the gastric tissues. It is surprising that the authors don't better define the tuft cell population.

Please note that Figure S2 has been moved to now be Figure S8.

We have consistently used DCLK1 staining as a hallmark for identification of murine tuft cells in IF and IHC as a widely accepted marker for tuft cells that is consistently used throughout a majority of published reports on tuft cells in mice (16, 17). However, rather than building on an intracellular marker (i.e. Dclk1), we have used dual stains for the extracellular markers SiglecF and Cd24 for flow cytometry, because the expression of these marker substantially overlap with Dclk1 expression and highly enrich for tuft cells as shown in by our scRNA expression analysis below (data for reviewer only). Meanwhile, we find that inclusion of additional staining for the IL25R subunit IL17rb, and/or the IL13 receptor, Il13ra, also stains a large majority of the Dclk1+SiglecF+Cd24+ tuft cells .

Comment 4. The definition of iILC2s and nILC2s raises concerns. In figure S2B the authors use an anti-IL-25R antibody to define iILC2s, however most of the experiments do not use this antibody to define these cells and instead cells are referred to as ST2-negative. It is not clear if these are ILC2s at all. Do the authors find that gastric ILC2s express IL-25R at homeostasis as reported elsewhere (PMID:32891625). The authors should look at single IL-25 and ST2 receptor-positive cells and IL-25R/ST2 double positive cells in their experiments. GATA3 staining should also be included. It is also not investigated whether these cells are local or recruited – to warrant being called iILC2s and nILC2s.

When we refer to ST2-negative CD45⁺KLRG1⁺CD90.2⁺Lineage⁻ cells, we are confident with our classification as ILC2s as there are no other known ST2-negative cell types that express these markers. We have consistently confirmed RNA expression of IL17Rb and GATA3 in our ST2-negative populations and found that all KLRG1⁺CD90.2⁺Lineage⁻CD45⁺ cells also express GATA3 (see below Panel A, for review only).

While we did not determine Il17Rb expression in WT ILC2s by flow cytometry, our single cell analysis revealed similar expression of Il17rb across our gastric tumor model and WT controls (Figure 3C, E). Importantly we were unable to detect ST2 expression on ILC2s outside of WT mice (Figure S8B) and find that iILC2s in our metaplasia and cancer models lack ST2 expression.

While some literature on ILC2s does specify whether iILC2s were recruited or occurred locally, most papers make no reference suggest that this population is recruited from outside the affected tissue.

This is most notable in studies that focus on the response of ILC2s to intestinal parasites, and suggest that intestinal iILC2 population migrates to the lungs, while remaining ILC2s in the intestine still retain the phenotype of iILC2s (18).

Comment 5. There are issues with the order of figures and parts of figures e.g. 3B is only cited after Figure 4 – this is not normal procedure.

We have rearranged the order of figures to be referred to in the text in ascending order.

Comment 6. The gp130^{Y757F/ Y757F} mutant knock-in mouse needs to be introduced. Especially as many readers may assume F refers to Flox.

The gp130^{Y757F/Y757F} mouse has previously been published (Tebbutt, Nat Med 2002). We now added the below relevant background to the gp130^{Y757F/Y757F} mice in the Results section. (PDF page 4, line 120 - 125)

“To do this we utilised the gp130^{Y757F/ Y757F} (gp130^{F/F}) mouse model, which spontaneously develop SPEM-associated intestinal-type gastric adenomas from 4 weeks of age (Fig. S3A, B) (19). These tumors develop due to the excessive activation of Stat3 in response to a tyrosine (Y) to phenylalanine (F) knock-in substitution mutation in the common IL6 family receptor gp130 preventing the binding of the negative regulator Suppressor of cytokine signaling (Socs3)(19).”

Comment 7. The authors need to perform flow cytometry in their models to define other immune cell types. It is essential that CD4+ Th1 and Th2 cells are identified and CD8+ T cells.

In order to address this query across all models investigated in our manuscript, we elected to perform IHC analysis for CD4 and CD8 in the mucosa and submucosa (see panel below, for review only), rather than flow cytometric analysis which we feel is outside of the scope of this work. As this assessment revealed no significant difference of abundance between CD4 or CD8 cells during metaplasia or tumor development, either during tuft cell ablation or ILC2 depletion, we did not perform further staining to differentiate Th1 and Th2 cells.

Panel A: HDTmx model (representative IHC and associated quantifications);

Panel B: MNU gastric adenocarcinoma (representative IHC and associated quantifications);

Panel C: Tuft cell ablation in gastric adenoma model;

Panel D: ILC2 ablation in gastric adenoma model;

Panel E: α-IL13 therapeutic intervention in gastric adenoma model;

Panel F: α-IL25 therapeutic intervention in gastric adenoma model)

Comment 8. This requirement is true also where the IL5-Tom-Cre mouse is used as the authors may also be depleting Th2 cells in the tissues or IL-5-producing myeloid cells.

While we cannot rule out minor depletion of other immune populations that express IL5, our staining for CD8 and CD4 T cells in the IL5-Tom-Cre model showed no reduction of staining between ILC2^{WT} and ILC2^{DTA} mice (please also refer to Panel D in the reply to comment 7). Because these observations strongly suggest that CD4 and CD8 T cells in the gastric mucosa are not depleted in our ILC2^{DTA} mice, reduction of tumor burden in this model is T cell independent. Indeed, activity of the IL5-Tom reporter confirmed that ILC2s make up more than 90% of all IL5 expressing cells in the stomach (see panel below, for review only), and therefore being the predominant cell population that is by the activity of IL5-Tom-Cre.

Comment 9. In most experiments there is no mention of the number of experimental repeats to validate that the results are robust.

We have added this information to the figure legend of all experiments.

Comment 10. In Figure 3A/3B there is no indication of the numbers counted.

We have added this information to the figure legend.

Comment 11. Fig 4B. The authors quotes percentages, but isn't this just a single data point when comparing WT with Tg (or are there more repeats?)?

We have clarified in the text that this dataset is made up of cells from n=10 of each gp130^{+/+} wild-type and gp130^{F/F} mutant mice. Please observe that this is now Figure 3B in the revised version of the manuscript.

Comment 12. Fig 4C- stain for IL-13R and IL-25R protein. These are tuft-like cells, but are induced by metaplasia, how similar are they to those in the small intestine?

As per our scRNA data presented in response to comment #3 by this reviewer, this analysis confirmed that most of the cells co-expressing SiglecF and Cd24 also expressed the classical tuft cell marker Dclk1, while a large subset of this population also expresses the IL13 receptor gene Il13ra.

When integrating expression for classical markers of intestinal tuft cells (i.e. Dclk1, Alox5, CD24, SiglecF and Trpm5) (16) with our data, we find that gastric tuft cells in Gp130^{+/+} and Gp130^{F/F} mice also express these genes (see below, for review only). However, across these tufts cells, we find differences in gene expression between tissues and conditions (i.e. intestine, normal gastric mucosa and metaplastic gastric mucosa), suggesting some functional plasticity between tuft cells derived from different organs and/or disease stages, in agreement with tissue-specific expression patterns of tuft cells previously reported across multiple organs (20-22). Importantly, we note that CD24 and SiglecF as tuft cell marker we use here belong those marker consistently expressed across intestinal and gastric tuft cells, and which have previously been used to enrich tuft cells in the intestine (16, 17).

Comment 13. In many experiments and figure legends it is unclear if experimental repeats have been performed (how many repeats and how many mice per group). These are essential to confirm the robustness of the data and author claims.

We are now consistently providing this information in each figure legend.

Comment 14. Only 3 data points in Fig 5E and no repeat.

We have performed additional experiments to increase the number data points for this experiment.

Comment 15. What are all the GATA3-positive T cells in the images? They greatly outnumber the ILC2. Are these Th2 cells? Are they IL-13 producers? Are they IL-5 producers? Are they deleted in the IL5DTA mice? If they make IL-13 then why isn't this IL-13 active on epithelial cells over a 4-month model?

While we have not directly identified these cells, they are likely to be T cells, being the only GATA3⁺ CD3⁺ cell population. While there is a significant population of these cells, we found that in our models ILC2s make up the majority of all gastric immune cells that express IL5 and thus CD3⁺GATA3⁺ cells are not being deleted in our IL5^{DTA} mice (please also refer to our responses above to comments 5 and 8). Additionally, we found there was no significant difference in T cell populations in our models using the IL5DTA mice (see our response to comment 7 raised by this reviewer). We have also shown that ILC2s are the primary source of IL13 across our models when comparing IL13 expression of sorted ILC2s to other immune cells (Figure 3E). Finally, we have also examined levels of IL13 across all of our models by ELISA and observed is a significant reduction in IL13 following ILC2s depletion (newly added data to Figure S5E). While not precluding that other immune cell populations may contribute to IL13 production, our observations provide a compelling argument that ILC2s are the predominant source for IL13 detected in our models.

Comment 16. Are the other T cells CD8 cytolytic cells?

While we have not formally assessed the cytolytic activity of CD8 cells in our models, we note that the frequency between CD4 and CD8 cells does not change during metaplasia and tumor development (see our response above to comment 7). Moreover, the presence of the gp130^{Y757F} mutation as a germline mutation, makes it unlikely that a T-cell mediated anti-tumor response would account for tumor control in this model consistent with our previous observations that the tumor burden remains comparable between 16-20 week-old gp130^{Y757F} and gp130^{Y757F};Rag1^{-/-} mice (23).

Comment 17. Figure 6A – this is confusing. Are the lines mislabelled? If not then isn't this contradictory with the main hypothesis?

We apologise for this oversight. The lines in this figure were mislabelled, we have corrected this.

Comment 18. PDF line 251 (previously was 215). The use of 'novel' is not justified given the existing number of examples where this axis has been identified.

While we believe this to be the first instance of the tuft cell – ILC2 cytokine circuit being identified during gastric cancer, we have removed the term 'novel' as we agree that the general concept of this circuit during parasitic wound-healing has previously been described.

Comment 19. The authors have to better cite and discuss the already reported role of IL-33 (Mayer et al 2020 PMID 32891625) as these have also been studied in SPEM models. There is significant overlap in the phenotypes. How do IL-25 and IL-33 work alongside one another? This relationship is of significant importance and needs more attentions here.

We now refer to the findings by Mayer et al 2020 more thorough throughout our manuscript. Specifically, we have expanded on the link between IL33, IL25 and ILC2s in both the Introduction and the Discussion. (PDF page 2, line 71 – 76; PDF page 7, line 242 – 249)

“Moreover, IL33 responsive ILC2s have been linked to Helicobacter pylori driven gastric metaplasia in humans and mice (1). Additionally, IL33 activated ILC2s were proposed as a source of IL13 during chemically induced metaplasia (6, 7), with depletion of ILC2s resulting in reduced tuft cell hyperplasia and gastric metaplasia (7). While ILC2s were found to be increased in the blood of gastric cancer patients (8), little else is known about their interactions with gastric tumor development.”

“Although the cytokine responsiveness of ILC2s is contextual, IL33 and IL25 remain the major drivers of activation (7, 24), with prominent roles for IL25 during skin allergies (25), pulmonary fibrosis (26) and helminth defense (27). Meanwhile, IL33 activated ILC2s play a major role in driving SPEM development through the recruitment and polarization of macrophages (7, 28, 29). Additionally, IL13 production has been linked to mast cells, B cells, T cells and macrophages during genetically induced gastric metaplasia, with inhibition of IL13 reducing metaplasia development (29). While these processes rely heavily on IL13-producing ILC2s, and a link between ILC2s and tuft cell abundance has been identified (7, 29), a clear role for IL25 has yet to be determined.”

While our experiments focus on the role of IL25-responsive cells as the most abundant population of ILC2s, we now speculate in the Discussion on the role of IL33 as an alternate initiator of this circuit in the less abundant IL33-responsive ILC2. (PDF page 8, line 275 – 285)

“Based on our data and current literature, we propose a central role for ILC2 as an inflammatory sentinel producing IL13 not only in response primarily to IL25, but also to local accumulation of the alarmin IL33, consistent with findings that genetic ablation of either IL33 or IL13 expression prevented the formation of experimentally-induced gastric metaplasia. In the latter case it is thought that IL33 is either released from damaged epithelium or secreted by metaplasia-associated macrophages (6, 7). Thus, we surmise that during parietal cell loss, ILC2s are initially activated by either epithelial cell damage or macrophage-derived IL33, which in turn induces tuft cell expansion via IL13 signaling in gastric chief and/or metaplastic SPEM cells (Fig. 6D). Once the tuft cell population has sufficiently expanded, we show here that IL25 becomes the dominant driver for the expansion and activation of tissue-resident mucosal iILC2 at the expense of the IL33 responsive nILC2s which do not expand under these conditions.”

Comment 20. PDF line 286 (previously was 257) – adenocarcinoma models were also used in these studies. Not Apcmin?

This has been updated to reflect that the study referred to the use of the Apc^{1322T/+} model and not Apc^{min} mice.

Comment 21. The authors do not undertake therapeutic treatment in adenocarcinoma ‘cancer’ models, but in pre-cancerous adenoma models. Could the authors use H pylori or TFF1-KO models to test cancer progression? I appreciate that the MNU model takes a year, but the ‘control’ of cancer is of significant interest.

While we have not undertaken therapeutic treatment on H. pylori infected mice, our HDTmx model reliably mimics metaplasia associated with chronic H. pylori infection (30-32), and we note that both, tuft cells and ILC2s are also increased in the gastric mucosa during H. pylori infections in mice (33, 34). Likewise, excessive transcriptional activity of Stat3, which also occurs in our gp130^{F/F} model use for HDTmx-induced metaplasia, has been shown as an absolute requirement for helicobacter-induced metaplasia (35). Accordingly, we surmise that therapeutic targeting of either tuft cells or ILC2s in a setting of chronic helicobacter infection, similarly to the setting of excessive Stat3 activity in the gp130^{F/F} model, is also likely to limit metaplasia akin to our findings in the HDTmx model. To improve clarity, we have expanded our explanation of the similarities between the HDTmx and helicobacter models in the main text. (PDF page 3, line 86 - 90)

“Intestinal metaplasia is the leading risk factor for GC, while increased abundance of tuft cells and ILC2s in the gastric mucosa has been associated with chronic H. pylori infection (33, 34) and metaplasia (7, 28, 29, 36). We therefore induced gastric metaplasia through administration of high dose tamoxifen (HDTmx; 250mg/kg) (30) which has been reported to induce a similar phenotype to that observed as a result of chronic Helicobacter infection (30-32)”

Our MNU+NaCl experiments demonstrated that loss of ILC2s and tuft cells reduces IL13 and IL25 expression, and as a consequence also reduces the number of MNU-induced tumors. This highlights the involvement of the tuft cell – ILC2 circuit as a driver of tumor growth across unrelated gastric cancer models and therefore strongly argues that either α -IL13 or α -IL25 treatments could provide therapeutic benefits. Follow-up studies that are beyond the scope of the present investigations will need to define the benefits and possible limitations of such therapies for human gastric cancer patients.

Comment 22. To confirm their hypothesis, the authors need to better link the cytokine and cell responses e.g. IL-25R or IL-13R deletion from tuft cells, IL-13 deletion from ILC2 etc, or perhaps use cell transfer models.

We agree that there is a great importance in the linking of IL25 and IL13 to tuft cells and ILC2s respectively. However, we believe our data provides compelling evidence to functionally link IL25 and IL13 signaling to tuft cell and ILC2 function:

- In our tuft cell and ILC2 ablation experiments, we observed a significant reduction in the abundance of these cytokines (see Figure 1 and 2), and this was associated with reduced metaplasia and tumor burden.*
- Ablation of tuft cells resulted in a significant reduction of ILC2, while ablation of ILC2 was associated with reduced abundance of tuft cells.*
- Single cell RNA analysis suggests that tuft cells are the predominant cell population expressing IL25, while ILC2 are the predominant cellular source for IL13 (Fig. 3 C-E).*
- Ablation of Dclk1+ tuft cells reduced IL25 expression, while conversely, ablation of ILC2 was associated with reduced IL13 expression.*
- Treatment with either α -IL13 or α -IL25 reduced the abundance of both, ILC2s and tuft cells (Fig. 5E-H) and this was accompanied by reduced expression of both IL13 and IL25 at the RNA and protein level.*

Minor comment 1. PDF line 141 (previously was 123) should read gp130F/F;ILC2WT

We have corrected this error.

Minor comment 2. PDF line 184 (previously was 170) refers to select subsets, but only ILC2 are shown. Were there supposed to be other cell types?

This description was due to correlating the single cell and qPCR data together. We have amended the sentence to now specifically refer to the data in Fig. 3E (PDF page 5, line 184 – 185)

“Additionally, we observed that Il13 was predominantly expressed in gp130^{F/F} ILC2s’.”

Minor comment 3. The term therapeutic vulnerabilities is confusing.

We have removed this phrase from the manuscript and replaced it with more descriptive statements. (PDF page 1, line 28-31. PDF page 6, line 196 – 198)

“Our findings suggest novel roles for ILC2 and tuft cells, along with their associated cytokine IL13 and IL25 as gatekeepers and enablers of metaplastic transformation and gastric tumorigenesis, thereby providing an opportunity to therapeutically inhibit early-stage gastric cancer through repurposing antibody-mediated therapies.”

“To explore whether these in vitro dependencies of gastric tumor organoids could translate to therapeutic approaches in vivo, we treated tumor-bearing gp130^{F/F} mice with α -IL13 or α -IL25 neutralizing antibodies (Fig. 5A).”

Minor comment 4. PDF line 291 (previously was 263). What do the authors mean by ‘phylogenetically optimized’? What is the evidence for this?

We have removed this phrase in the interest of clarity and reworded the sentence. (PDF page 8, line 291 – 293)

“Our study demonstrates that the tuft cell-ILC2 feed-forward circuit, originally identified as a repair mechanism of the intestinal epithelium during helminth infection, provides another facet of wound-healing mechanism being hijacked to promote progression from neoplastic transformed cells.”

Minor comment 5. Fig 4A labelling too small.

We have increased the size of Figure 4A labelling.

Minor comment 6. Fig5A – a-IL-25, change to anti or alpha as used elsewhere.

This has been updated.

Minor comment 7. Fig s6G/H mentioned after s7A/B.

We have updated the order of all figures to correct this.

Response to specific comments raised by **Reviewer #3**:

Reviewer Summary: “This study proposes that pro-tumorigenic properties caused by the crosstalk between intestinal tuft cells and type 2 innate lymphoid cells (ILC2) may be promoting gastric metaplasia/tumorigenesis. Strengths of study include the use of multiple mouse models to investigate the role of ILC2s and tuft cells in gastric metaplasia/adenomas. These include the use of tamoxifen induced metaplasia in control and tuft cell deficient mice, tumor development in GP130F/F mutant mice crossed to tuft cell deficient as well as ILC2 deficient mice as well as cytokine neutralizing antibodies. The data clearly show that tumor mass is reduced in GP130F/F when tuft cells and ILC2s are deficient, and when mice are treated with IL-13 or IL-25 neutralizing antibodies. These data support a role for tuft cells and ILC2s in regulating tumor growth in these models. These results are noteworthy.”

Comment 1. One is the lack of distinction between SPEM, which refers to the presence of MUC6 and TFF2 expressing metaplastic cells that share features with mucous producing deep antral glands from the antrum of the stomach, and intestinal metaplasia, which share features with MUC2/TFF3 intestinal type goblet cells. These are distinct types of gastric metaplasia and this study does not distinguish between the two (Figure 1). It is difficult to determine how metaplasia is defined and quantified in Figure 1. In this model metaplasia (SPEM) is usually measured by TFF2/GIF/MUC6 expressing cells at the base of glands because TFF2 is also expressed in other cell types (neck cells, pit cells).

The high-dose tamoxifen model of gastric metaplasia is an established model (30, 31, 37-40) for the rapid, and reversible, induction of spasmodic polypeptide-expressing (SPEM) metaplasia, with one of the defining characteristics being the loss of parietal cells. While we have originally used the latter feature to determine whether the gastric glands underwent SPEM (original Figure 1B, S1B), we now supplement these observations with fluorescent imaging and quantification of GIF/GS-II/TFF2 expression as suggested by the reviewer as being the contemporary gold-standard to confirm SPEM (updated Figure 1B, S1B).

Comment 2. Much of the cytokine and cytokine receptor expression data are correlative (mRNA expression), and only one IL-13 receptor is examined (Il13ra but not Il4ra). Thus, the proposed Tuft cell (IL25) – ILC2 (IL13) circuit, which is the unique aspect of this study, is not clearly established.

We have expanded our scRNA analysis to include Il4ra as the second subunit of the IL13 receptor (Fig. 3C) and we have performed extensive ELISA analysis (Fig. S5) to substantiate our transcriptional analysis with protein data. Collectively, this data clearly demonstrates that tuft cells are the major cell type which co-expresses both subunits required for a functional IL13 receptor complex, namely Il13ra and Il4ra. We also confirm previous observations (41) that transcripts for both subunits could also be detected in granulocytic myeloid-derived suppressor cells (MDSCs) and therefore speculate that MDSCs may play a role in the initiation of the tuft cell-ILC2 circuit (41).

Comment 3. Other studies have demonstrated roles for ILC2s and IL-13 in tuft cell hyperplasia and gastric metaplasia (most were cited, not PMID: 34587523), so clearly establishing the Tuft cell (IL25) – ILC2 (IL13) circuit would distinguish this study from others. Perhaps demonstrating that IL25 production specifically by tuft cells or IL-13 production specifically by ILC2s enhance gastric metaplasia/tumorigenesis.

We now refer to the suggested study in the Discussion section, but note that this publication focuses on the role of IL13 during gastric metaplasia, but does neither mention tuft cells nor IL25 at all, which is one of the key points of our study. (PDF page 7, line 242 – 249)

“Although the cytokine responsiveness of ILC2s is contextual, IL33 and IL25 remain the major drivers of activation (7, 24), with prominent roles for IL25 during skin allergies (25), pulmonary fibrosis (26) and helminth defense (27). Meanwhile, IL33 activated ILC2s play a major role in driving SPEM development through the recruitment and polarization of macrophages (7, 28, 29). Additionally, IL13 production has been linked to mast cells, B cells, T cells and macrophages during genetically induced gastric metaplasia, with inhibition of IL13 reducing metaplasia development (29). While these processes rely heavily on IL13-producing ILC2s, and a link between ILC2s and tuft cell abundance has been identified (7, 29), a clear role for IL25 has yet to be determined.”

We also refer Reviewer 3 to our response to a similar question by Reviewer 2, Question 22.

Comment 4. Also, the human datasets tuft cell signatures and ILC2 signatures correlate with survival curves in patients with intestinal type gastric cancer, but it is not clear why lower signature scores (in red) have less overall survival. This needs some explanation.

We have updated the figure to correct the mis-labelled survival analysis. This figure now reflects the survival data appropriately (Fig. 6A).

Comment 5. Finally, the organoid experiments are presented in Figure 3, but discussed after Figure 4 in the results. It is not clear what are the sources of IL-25 (are there tuft cells in these organoids) that reduce growth in anti-IL25 treated organoids, or which cell types are responding to IL-25 when it increases the size of organoids. Some discussion would be helpful.

We have updated and corrected the order of figures throughout the manuscript.

While we cannot formally exclude that there may be traces of IL25 in the serum (or other constituents of the organoid growth medium), the most likely reason for the activity of the anti-IL25 antibody is the described tuft cell-IL25 autocrine signaling loop (42, 43). (PDF page 7, line 266 - 269)

“Additionally, reduced organoid growth observed in anti-IL25 treated cultures is consistent with the previous identification of tuft cells in gastric organoids (9), and highlights the possibility of tuft cell-derived IL25 as an autocrine-acting growth factor given the relative absence of IL25 receptor expression in other epithelial cell types.”

References

1. R. Li *et al.*, Group 2 Innate Lymphoid Cells Are Involved in Skewed Type 2 Immunity of Gastric Diseases Induced by Helicobacter pylori Infection. *Mediators Inflamm* **2017**, 4927964 (2017).
2. G. Ercolano, M. Falquet, G. Vanoni, S. Trabanelli, C. Jandus, ILC2s: New Actors in Tumor Immunity. *Front Immunol* **10**, 2801 (2019).
3. N. Jacquelot *et al.*, Blockade of the co-inhibitory molecule PD-1 unleashes ILC2-dependent antitumor immunity in melanoma. *Nat Immunol* **22**, 851-864 (2021).
4. J. A. Moral *et al.*, ILC2s amplify PD-1 blockade by activating tissue-specific cancer immunity. *Nature* **579**, 130-135 (2020).
5. M. J. Schuijs *et al.*, ILC2-driven innate immune checkpoint mechanism antagonizes NK cell antimetastatic function in the lung. *Nature immunology* **21**, 998-1009 (2020).
6. C. P. Petersen *et al.*, A signalling cascade of IL-33 to IL-13 regulates metaplasia in the mouse stomach. *Gut* **67**, 805-817 (2018).
7. A. R. Meyer *et al.*, Group 2 Innate Lymphoid Cells Coordinate Damage Response in the Stomach. *Gastroenterology* **159**, 2077-2091.e2078 (2020).
8. Q. Bie *et al.*, Polarization of ILC2s in peripheral blood might contribute to immunosuppressive microenvironment in patients with gastric cancer. *J Immunol Res* **2014**, 923135 (2014).
9. C. B. Westphalen *et al.*, Long-lived intestinal tuft cells serve as colon cancer-initiating cells. *The Journal of clinical investigation* **124**, 1283-1295 (2014).
10. M. Giannakis *et al.*, Molecular properties of adult mouse gastric and intestinal epithelial progenitors in their niches. *J Biol Chem* **281**, 11292-11300 (2006).
11. M. Middelhoff *et al.*, Dclk1-expressing tuft cells: critical modulators of the intestinal niche? *Am J Physiol Gastrointest Liver Physiol* **313**, G285-G299 (2017).
12. F. Gerbe, B. Brulin, L. Makrini, C. Legraverend, P. Jay, DCAMKL-1 expression identifies Tuft cells rather than stem cells in the adult mouse intestinal epithelium. *Gastroenterology* **137**, 2179-2180; author reply 2180-2171 (2009).
13. P. Chandrakesan *et al.*, Dclk1+ small intestinal epithelial tuft cells display the hallmarks of quiescence and self-renewal. *Oncotarget* **6**, 30876-30886 (2015).
14. R. May *et al.*, Identification of a novel putative gastrointestinal stem cell and adenoma stem cell marker, doublecortin and CaM kinase-like-1, following radiation injury and in adenomatous polyposis coli/multiple intestinal neoplasia mice. *Stem cells (Dayton, Ohio)* **26**, 630-637 (2008).
15. X. Wu, D. Qu, N. Weygant, J. Peng, C. W. Houchen, Cancer Stem Cell Marker DCLK1 Correlates with Tumorigenic Immune Infiltrates in the Colon and Gastric Adenocarcinoma Microenvironments. *Cancers (Basel)* **12**, (2020).
16. A. L. Haber *et al.*, A single-cell survey of the small intestinal epithelium. *Nature* **551**, 333-339 (2017).
17. C. Schneider *et al.*, A Metabolite-Triggered Tuft Cell-ILC2 Circuit Drives Small Intestinal Remodeling. *Cell* **174**, 271-284 e214 (2018).
18. Y. Huang *et al.*, S1P-dependent interorgan trafficking of group 2 innate lymphoid cells supports host defense. *Science* **359**, 114-119 (2018).
19. M. Ernst *et al.*, STAT3 and STAT1 mediate IL-11-dependent and inflammation-associated gastric tumorigenesis in gp130 receptor mutant mice. *The Journal of clinical investigation* **118**, 1727-1738 (2008).
20. L. Li, M. Ma, T. Duan, X. Sui, The critical roles and therapeutic implications of tuft cells in cancer. *Front Pharmacol* **13**, 1047188 (2022).
21. S. Nevo, N. Kadouri, J. Abramson, Tuft cells: From the mucosa to the thymus. *Immunol Lett* **210**, 1-9 (2019).
22. H.-A. Ting, J. Von Moltke, The Immune Function of Tuft Cells at Gut Mucosal Surfaces and Beyond. *The Journal of Immunology* **202**, 1321-1329 (2019).

23. M. Howlett *et al.*, Differential regulation of gastric tumor growth by cytokines that signal exclusively through the coreceptor gp130. *Gastroenterology* **129**, 1005-1018 (2005).
24. H. Takatori, S. Makita, T. Ito, A. Matsuki, H. Nakajima, Regulatory Mechanisms of IL-33-ST2-Mediated Allergic Inflammation. *Frontiers in immunology* **9**, 2004-2004 (2018).
25. J. M. Leyva-Castillo *et al.*, ILC2 activation by keratinocyte-derived IL-25 drives IL-13 production at sites of allergic skin inflammation. *Journal of Allergy and Clinical Immunology* **145**, 1606-1614.e1604 (2020).
26. E. Hams *et al.*, IL-25 and type 2 innate lymphoid cells induce pulmonary fibrosis. *Proceedings of the National Academy of Sciences* **111**, 367-372 (2014).
27. Y. Huang *et al.*, Inflammatory ILC2: An IL-25-activated circulating ILC population with a protective role during helminthic infection. *The Journal of Immunology* **198**, 68.13-68.13 (2017).
28. E. Choi *et al.*, Dynamic expansion of gastric mucosal doublecortin-like kinase 1-expressing cells in response to parietal cell loss is regulated by gastrin. *Am J Pathol* **185**, 2219-2231 (2015).
29. J. T. Busada *et al.*, Glucocorticoids and Androgens Protect From Gastric Metaplasia by Suppressing Group 2 Innate Lymphoid Cell Activation. *Gastroenterology* **161**, 637-652.e634 (2021).
30. W. J. Huh *et al.*, Tamoxifen induces rapid, reversible atrophy, and metaplasia in mouse stomach. *Gastroenterology* **142**, 21-24.e27 (2012).
31. J. B. Saenz, J. Burclaff, J. C. Mills, Modeling Murine Gastric Metaplasia Through Tamoxifen-Induced Acute Parietal Cell Loss. *Methods Mol Biol* **1422**, 329-339 (2016).
32. C. P. Petersen, J. C. Mills, J. R. Goldenring, Murine Models of Gastric Corpus Preneoplasia. *Cellular and Molecular Gastroenterology and Hepatology* **3**, 11-26 (2017).
33. C. Pinzon-Guzman *et al.*, Evaluation of Lineage Changes in the Gastric Mucosa Following Infection With *Helicobacter pylori* and Specified Intestinal Flora in INS-GAS Mice. *J Histochem Cytochem* **67**, 53-63 (2019).
34. N. Satoh-Takayama *et al.*, Bacteria-Induced Group 2 Innate Lymphoid Cells in the Stomach Provide Immune Protection through Induction of IgA. *Immunity* **52**, 635-649 e634 (2020).
35. J. J. Balic *et al.*, Constitutive STAT3 Serine Phosphorylation Promotes *Helicobacter*-Mediated Gastric Disease. *The American Journal of Pathology* **190**, 1256-1270 (2020).
36. M. Saqui-Salces *et al.*, Gastric tuft cells express DCLK1 and are expanded in hyperplasia. *Histochemistry and cell biology* **136**, 191-204 (2011).
37. J. Burclaff, L. H. Osaki, D. Liu, J. R. Goldenring, J. C. Mills, Targeted Apoptosis of Parietal Cells Is Insufficient to Induce Metaplasia in Stomach. *Gastroenterology* **152**, 762-766 e767 (2017).
38. J. Matsuo *et al.*, Identification of Stem Cells in the Epithelium of the Stomach Corpus and Antrum of Mice. *Gastroenterology* **152**, 218-231.e214 (2017).
39. K. A. Bockerstett *et al.*, Single-cell transcriptional analyses of spasmolytic polypeptide-expressing metaplasia arising from acute drug injury and chronic inflammation in the stomach. *Gut* **69**, 1027-1038 (2020).
40. C. Lee, H. Lee, S. Y. Hwang, C. M. Moon, S. N. Hong, IL-10 Plays a Pivotal Role in Tamoxifen-Induced Spasmolytic Polypeptide-Expressing Metaplasia in Gastric Mucosa. *Gut Liver* **11**, 789-797 (2017).
41. E. Jou *et al.*, An innate IL-25-ILC2-MDSC axis creates a cancer-permissive microenvironment for *Apc* mutation-driven intestinal tumorigenesis. *Sci Immunol* **7**, eabn0175 (2022).
42. C. Bezencon *et al.*, Murine intestinal cells expressing *Trpm5* are mostly brush cells and express markers of neuronal and inflammatory cells. *The Journal of comparative neurology* **509**, 514-525 (2008).
43. J. von Moltke, M. Ji, H. E. Liang, R. M. Locksley, Tuft-cell-derived IL-25 regulates an intestinal ILC2-epithelial response circuit. *Nature* **529**, 221-225 (2016).

REVIEWERS' COMMENTS

Reviewer #2 (Remarks to the Author):

The revised manuscript is considerably improved and now includes a greater proportion of relevant citations of already published studies. The manuscript is now more intelligible and the data better presented. The authors have made a strong case for the role of ILC2s and Tuft cells in gastric cancer - similar to those proposed in other cancers of the gastrointestinal tract. These data will be of significant interest in the field.

While the focus on ILCs is warranted, the authors have largely ignored other immune cells which may play potential roles, e.g. T cells. In response to this criticism they have provided immunohistology of T cells to the reviewers. However, this is not an adequate analysis of T cell subsets and does not probe the potential cellular function of these cells e.g. cytokine production, lytic activity, receptor expression. Given the limitations of almost all cell-depletion models and the absence of experiments to categorically exclude a possible role for T cells (unless they have other data for this) the authors should include a statement to acknowledge that they cannot formally exclude a role for T cells.

Reviewer #3 (Remarks to the Author):

This revised manuscript presents data to support a role for ILC2s, tuft cells, IL13, and IL25 in gastric epithelial cell changes associated with preneoplastic lesions (metaplasia) and tumor formation in mouse models. While there are reports that IL13 and ILC2s are involved in gastric metaplasia, this is the first to propose the importance of cross talk between tuft cells and ILC2s in promoting gastric metaplasia. Some correlative data from human datasets support tuft cell and ILC2 expression signatures associated with poorer survival in patients with intestinal-type GC. The strengths of the study include the significance, gastric cancer is a major cause of cancer related deaths and gastric carcinogenesis is poorly understood from an immunological perspective. The study also uses several models to test the hypothesis, including acute injury models, longer term tumor models, scRNA sequencing, and analyses of human datasets. The data supporting a role for tuft cells and IL25 in gastric metaplasia is significant and distinguishes this from other studies.

The authors did a nice job of addressing most of the concerns/suggestions in the previous reviews were addressed in this revised version, and this manuscript is improved.

Two issues that could be clarified.

First, the authors open up the results section mentioning "intestinal metaplasia" lines 86- "Intestinal metaplasia is the leading risk factor for GC, while increased abundance of tuft cells and ILC2s in the gastric mucosa has been associated with H. pylori infection (47, 48) and metaplasia (43, 49-51)." They then go on to examine metaplasia in an acute mouse model using high dose tamoxifen to injure the gastric mucosa. This "SPEM" is considered pseudopyloric metaplasia, which is antralization of the corpus region of the stomach, which is distinct from intestinal metaplasia, which describes gastric glands taking on intestinal features. I would suggest clarifying or dropping the comparison of intestinal metaplasia when using the mouse model of pyloric or pseudopyloric metaplasia (SPEM).

In this same figure (Figure 1B) the authors have responded to the first critiques by staining for GIF and GSII to identify metaplasia (SPEM) as co-staining for GIF/MUC6/TFF\2. It is not clear why there is so little GSII staining in the neck regions of the corpus glands, as GSII is normally expressed at high levels by mucous neck cells in healthy glands. This could be a staining issue, an imaging issue, or some other issue. Could the authors address whether they see neck cells staining for GSII and if so, include images that clearly show this staining?

We thank all the reviewers for their additional comments. Please find below our detailed responses to the issue raised by the reviewer for our manuscript *"A tuft cell - ILC2 signaling circuit provides therapeutic targets to inhibit gastric metaplasia and tumor development"*.

Reviewer #2

Remarks to the Author: "The revised manuscript is considerably improved and now includes a greater proportion of relevant citations of already published studies. The manuscript is now more intelligible and the data better presented. The authors have made a strong case for the role of ILC2s and Tuft cells in gastric cancer - similar to those proposed in other cancers of the gastrointestinal tract. These data will be of significant interest in the field.

While the focus on ILCs is warranted, the authors have largely ignored other immune cells which may play potential roles, e.g. T cells. In response to this criticism they have provided immunohistology of T cells to the reviewers. However, this is not an adequate analysis of T cell subsets and does not probe the potential cellular function of these cells e.g. cytokine production, lytic activity, receptor expression. Given the limitations of almost all cell-depletion models and the absence of experiments to categorically exclude a possible role for T cells (unless they have other data for this) the authors should include a statement to acknowledge that they cannot formally exclude a role for T cells."

We have updated Fig. 6D to include T cells as possibly playing a role in the gastric tuft cell-ILC2 circuit. We have also added a paragraph to the discussion addressing the potential role T cells could play. (PDF page 8, line 297 - 301)

"Additionally, T cells have previously been shown to work with ILC2s in the clearance of helminth infections, where mice that lacked IL13 producing T cells had a reduced ILC2 response and helminth clearance (37). Because of the limited antigenicity of the gastric tumors in gp130^{F/F} mice and our inability to detect IL13- or IL25-responsive MDSCs in these tumors, we cannot exclude an indirect contribution of IL25 signaling via MDSCs or IL13 produced by T cells to GC development."

Reviewer #3

Remarks to the Author: This revised manuscript presents data to support a role for ILC2s, tuft cells, IL13, and IL25 in gastric epithelial cell changes associated with preneoplastic lesions (metaplasia) and tumor formation in mouse models. While there are reports that IL13 and ILC2s are involved in gastric metaplasia, this is the first to propose the importance of cross talk between tuft cells and ILC2s in promoting gastric metaplasia. Some correlative data from human datasets support tuft cell and ILC2 expression signatures associated with poorer survival in patients with intestinal-type GC. The strengths of the study include the significance, gastric cancer is a major cause of cancer related deaths and gastric carcinogenesis is poorly understood from an immunological perspective. The study also uses several models to test the hypothesis, including acute injury models, longer term tumor models, scRNA sequencing, and analyses of human datasets. The data supporting a role for tuft cells and IL25 in gastric metaplasia is significant and distinguishes this from other studies.

The authors did a nice job of addressing most of the concerns/suggestions in the previous reviews were addressed in this revised version, and this manuscript is improved.

Two issues that could be clarified.

First, the authors open up the results section mentioning "intestinal metaplasia" lines 86- "Intestinal metaplasia is the leading risk factor for GC, while increased abundance of tuft cells and ILC2s in the gastric mucosa has been associated with H. pylori infection (47, 48) and metaplasia (43, 49-51)." They then go on to examine metaplasia in an acute mouse model using high dose tamoxifen to injure the gastric mucosa. This "SPEM" is considered pseudopyloric metaplasia, which is antralization of the corpus region of the stomach, which is distinct from intestinal metaplasia, which describes gastric glands taking on intestinal features. I would suggest clarifying or dropping the comparison of intestinal metaplasia when using the mouse model of pyloric or pseudopyloric metaplasia (SPEM).

We have removed the mention of Intestinal metaplasia from the manuscript, instead focussing on metaplasia, to better reflect that SPEM is inducing metaplasia and not intestinal metaplasia (PDF page 3, line 92).

In this same figure (Figure 1B) the authors have responded to the first critiques by staining for GIF and GSII to identify metaplasia (SPEM) as co-staining for GIF/MUC6/TFF\2. It is not clear why there is so little GSII staining in the neck regions of the corpus glands, as GSII is normally expressed at high levels by mucous neck cells in healthy glands. This could be a staining issue, an imaging issue, or some other issue. Could the authors address whether they see neck cells staining for GSII and if so, include images that clearly show this staining?

We have updated the images in Figure 1b and Figure S1b to ensure they reflect the results and clearly show the GSII staining neck cells.